# DMesh: A Differentiable Mesh Representation

**Sanghyun Son[1]  Matheus Gadelha[2]  Yang Zhou[2]  Zexiang Xu[2]  Ming C. Lin[1]  Yi Zhou[2]**
[1]University of Maryland, College Park, {shh1295,lin}@umd.edu
[2]Adobe Research, {gadelha,yazhou,zexu,yizho}@adobe.com

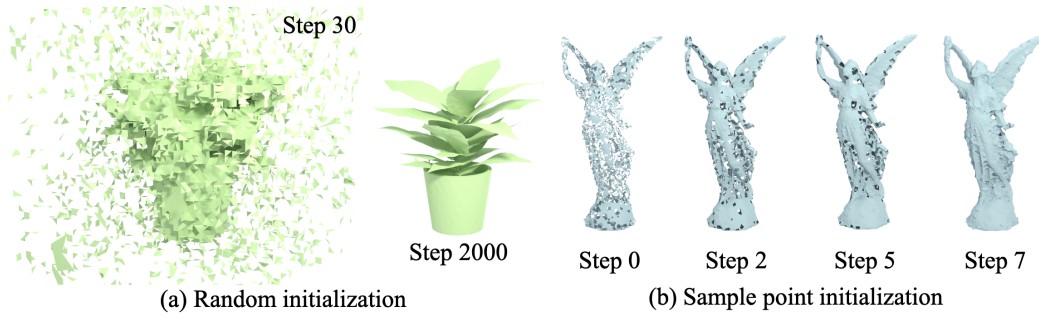

Figure 1: ($\rightarrow$) **Optimization process.** We can optimize our mesh starting from either (a) random state or (b) initialization based on sample points for faster convergence. Mesh connectivity changes dynamically during the optimization. To make this topology change possible, we compute existence probability for an arbitrary set of faces in a differentiable manner.

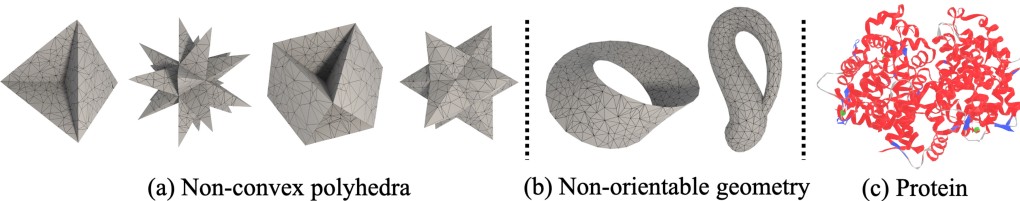

Figure 2: **Versatility of DMesh.** DMesh can represent diverse geometry in differentiable manner, including (a) non-convex polyhedra of different Euler characteristics, (b) non-orientable geometries (Möbius strip, Klein bottle), and (c) complex protein structure (colored for aesthetic purpose).

## Abstract

We present a differentiable representation, **DMesh**, for general 3D triangular meshes. **DMesh** considers both the geometry and connectivity information of a mesh. In our design, we first get a set of convex tetrahedra that compactly tessellates the domain based on Weighted Delaunay Triangulation (WDT), and select triangular faces on the tetrahedra to define the final mesh. We formulate probability of faces to exist on the actual surface in a differentiable manner based on the WDT. This enables **DMesh** to represent meshes of various topology in a differentiable way, and allows us to reconstruct the mesh under various observations, such as point clouds and multi-view images using gradient-based optimization. We publicize the source code and supplementary material at our project page [1].

---

[1]https://sonsang.github.io/dmesh-project

38th Conference on Neural Information Processing Systems (NeurIPS 2024).

# 1 Introduction

Polygonal meshes are widely used in modeling and animation due to their diverse, compact and explicit configuration. Recent AI progress has spurred efforts to integrate mesh generation into machine learning, but challenges like varying topology hinder suitable differentiable mesh representations. This limitation leads to reliance on differentiable intermediates like implicit functions, and subsequent iso-surface extraction for mesh creation (Liao u. a., 2018; Guillard u. a., 2021; Munkberg u. a., 2022; Shen u. a., 2023, 2021; Liu u. a., 2023b). However, meshes generated by such approaches can be misaligned at sharp regions and unnecessarily dense (Shen u. a., 2023), not suitable for down-stream applications that require light-weight meshes. This limitation necessitates us to develop a truly differentiable mesh representation, not the intermediate forms.

The fundamental challenge in creating a differentiable mesh representation lies in formulating both the vertices' geometric features and their connectivity, defined as edges and faces, in a differentiable way. Given a vertex set, predicting their connectivity in a free-form way using existing machine learning data-structures can cost significant amount of computation and be difficult to avoid irregular and intersecting faces. Consequently, most studies on differentiable meshes simplify the task by using a mesh with a pre-determined topology and modifying it through various operations (Zhou u. a., 2020; Hanocka u. a., 2019; Palfinger, 2022; Nicolet u. a., 2021). This work, on the contrary, ambitiously aims to establish a general 3D mesh representation, named as DMesh, where both mesh topology and geometric features (e.g. encoded in vertex location) can be simultaneously optimized through gradient-based techniques.

Our core insight is to use differentiable Weighted Delaunay Triangulation (WDT) to divide a convex domain, akin to amber encapsulating a surface mesh, into tetrahedra to form a mesh. To create a mesh with arbitrary topology, we select only a subset of triangular faces from the tetrahedra, termed the "real part", as our final mesh. The other faces, the "imaginary part", support the real part but are not part of the final mesh (Figure 4). We introduce a method to assess the probability of a face being part of the mesh based on weighted points that carry positional and inclusiveness information. Optimization is then focused on the points' features to generate the triangular mesh. The probability determination allows us to compute geometric losses and rendering losses during gradient-based optimization that optimizes **connectivity and positioning**.

The key contributions of our work can be summarized as follows.

- We present a novel differentiable mesh representation, **DMesh**, which is versatile to accommodate various types of mesh (Figure 2). The generated meshes can represent shapes more effectively, with much less number of vertices and faces (Table 2).

- We propose a computationally efficient approach to differentiable WDT, which produces robust probability estimations. While exhaustive approach (Rakotosaona u. a., 2021) requires quadratic computational cost, our method runs in approximately *linear time*.

- We provide efficient algorithms for reconstructing surfaces from both point clouds and multi-view images using DMesh as an intermediate representation.

- We finally propose an effective regularization term which can be used for mesh simplification and enhancing triangle quality.

Additionally, to further accelerate the algorithm, we implemented our main algorithm and differentiable renderer in CUDA, which is made available for further research.

# 2 Related Work

## 2.1 Shape Representations for Optimization

Recently, using neural implicit functions for shape representation gained popularity in graphics and vision applications (Mildenhall u. a., 2021; Zhang u. a., 2020; Liu u. a., 2020; Chen u. a., 2022a, 2023; Wang u. a., 2021; Yariv u. a., 2020). They mainly use volume density, inspired by (Mildenhall u. a., 2021), to represent a shape. However, because of its *limited accuracy* in 3D surface representation, neural signed distance functions (SDFs) (Yariv u. a., 2021; Wang u. a., 2021, 2023; Oechsle u. a., 2021) or unsigned distance functions (UDFs) (Liu u. a., 2023a; Long u. a., 2023) are often preferred.

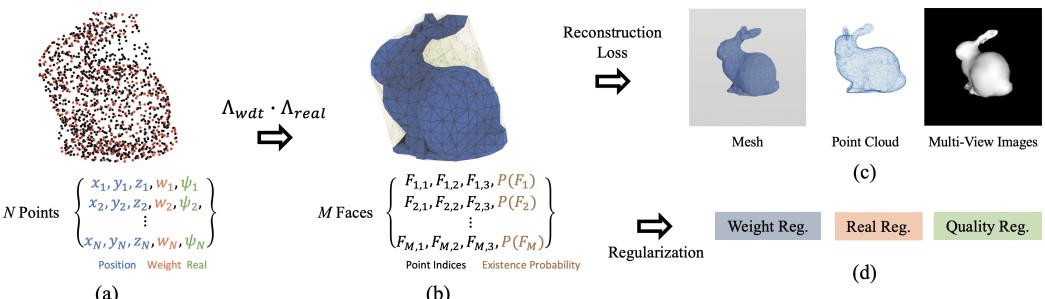

Figure 3: Our overall framework to optimize mesh according to the given observations. **(a)**: Each point is defined by a 5-dimensional feature vector: position, weight, and real value. Points with larger real values are rendered in red. **(b)**: Given a set of points, we gather possibly existing faces in the mesh and evaluate their probability in differentiable manner. **(c)**: We can compute reconstruction loss based on given observations, such as mesh, point cloud, or multi-view images. **(d)**: To facilitate the optimization process and enhance the mesh quality, we can use additional regularizations.

After optimization, one can recover meshes using iso-surface extraction techniques (Lorensen und Cline, 1998; Ju u. a., 2002).

Differing from neural representations, another class of methods directly produce meshes and optimize them. However, they assume that the overall mesh topology is fixed (Chen u. a., 2019; Nicolet u. a., 2021; Liu u. a., 2019; Laine u. a., 2020), only allowing local connectivity changes through remeshing (Palfinger, 2022). Learning-based approaches like BSP-Net (Chen u. a., 2020) allow topological variation, but their meshing process is not differentiable. Recently, differentiable iso-surface extraction techniques have been developed, resulting in high-quality geometry reconstruction of various topologies (Liao u. a., 2018; Shen u. a., 2021, 2023; Wei u. a., 2023; Munkberg u. a., 2022; Liu u. a., 2023b; Mehta u. a., 2022). Unfortunately, meshes relying on iso-surface extraction algorithms (Lorensen und Cline, 1998; Ju u. a., 2002) often result in unnecessarily dense meshes that could contain geometric errors. In contrast, **our approach addresses these issues: we explicitly define faces and their existence probabilities, and devise regularizations that yield simplified, but accurate meshes based on them** (Table 2). See Table 3 for more detailed comparisons to these other methods.

### 2.2 Delaunay Triangulation for Geometry Processing

Delaunay Triangulation (DT) (Aurenhammer u. a., 2013) has been proven to be useful for reconstructing shapes from unorganized point sets. It's been shown that DT of dense samples on a smooth 2D curve includes the curve within its edges (Brandt und Algazi, 1992; Amenta u. a., 1998a). This idea of using DT to approximate shape has been successfully extended to 3D, to reconstruct three-dimensional shapes (Amenta u. a., 1998b) for point sets that satisfy certain constraints. However, these approaches are deterministic. Our method can be considered as a differentiable version of these approaches, which admits gradient-based optimization.

More recently, Rakotosaona u. a. (2021) focused on this DT's property to connect points and tessellate the domain, and proposed a differentiable WDT algorithm to compute smooth inclusion, namely existence score of 2-simplices (triangles) in 2 dimensional space. However, it is not suitable to apply this approach to our 3D case, as there are computational challenges (Section 3.2). Other related work, VoroMesh (Maruani u. a., 2023), also used Voronoi diagrams in point cloud reconstruction, but their formulation cannot represent open surfaces and is only confined to handle point clouds.

## 3 Preliminary

### 3.1 Probabilistic Approach to Mesh Connectivity

To define a traditional, non-differentiable mesh, we specify the vertices and their connectivity. This connectivity is discrete, meaning for any given triplet of vertices, we check if they form a face in the mesh, returning 1 if they do and 0 otherwise. To overcome this discreteness, we propose a

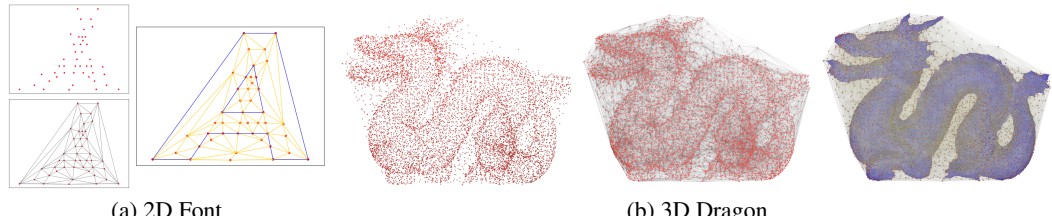

(a) 2D Font            (b) 3D Dragon

Figure 4: Illustration of our mesh representation for 2D and 3D cases. **(a):** Our representation in 2D for a letter "A". **(b):** Our representation in 3D for a dragon model. Blue faces are *"real part"* and yellow ones are *"imaginary part"*.

**probabilistic** approach to create a fully differentiable mesh – given a triplet of vertices, we evaluate the probability of a face existing. This formulation enables differentiability not only of **vertex positions** but also of their **connectivity**.

Note that we need a procedure that tells us the existence probability of any given face to realize this probabilistic approach. This procedure must be **1)** differentiable, **2)** computationally efficient, and **3)** maintain desirable mesh properties, such as avoiding non-manifoldness and self-intersections, when determining the face probabilities.

Among many possible options, we use *Weighted Delaunay Triangulation (WDT)* (Figure 6(a)) and a point-wise feature called the "real" value ($\psi$) to define our procedure. Each vertex in our framework is represented as a 5-dimensional[2] vector including position (3), WDT weight (1), and real value (1) (Figure 3(a)). Given the precomputed WDT based on vertices' positions and weights, we check the face existence probability of each possible triplets. Specifically, **1)** a face $F$ must exist in the WDT, and then **2)** satisfy a condition on the real values of its vertices to exist on the actual surface. We describe the probability functions for these conditions as $\Lambda_{wdt}$ and $\Lambda_{real}$:

$$\Lambda_{wdt}(F) = P(F \in \text{WDT}), \quad \Lambda_{real}(F) = P(F \in \text{Mesh} \mid F \in \text{WDT}). \tag{1}$$

Then we get the final existence probability function, which can be used in downstream applications (Figure 3), as follows:

$$\Lambda(F) = P(F \in \text{Mesh}) = \Lambda_{wdt}(F) \cdot \Lambda_{real}(F). \tag{2}$$

This formulation attains one nice property in determining the final mesh – that is, it prohibits self-intersections between faces. When it comes to the other two criteria about this procedure, $\Lambda_{wdt}$ function's differentiability and efficiency is crucial, as we design $\Lambda_{real}$ to be a very efficient differentiable function based on real values (Section 4.2). Thus, we first introduce how we can evaluate $\Lambda_{wdt}$ in a differentiable and efficient manner, which is one of our main contributions.

### 3.2 Basic Principles

To begin with, we use $(d, k)$ pair to denote a $k$-simplex ($\Delta^k$) in $d$-dimensional space. For a 3D mesh, we observe that our face $F$ corresponds to $(d = 3, k = 2)$ in Figure 5(b). To compute the probability $\Lambda_{wdt}$ for the face, we use power diagram (PD), which is the dual structure of WDT (Figure 6(a)). While previously Rakotosaona et al. (2021) proposed a differentiable 2D triangulation method for the $(d = k = 2)$ case (Figure 5(c)), it suffers from quadratically increasing computational cost (e.g. it takes 4.3 seconds to process $10K$ points in 2D, Table 4) and unreliable estimation when $(k < d)$. We will discuss later how our formulation conquer these computational challenges. Our setting for 3D meshes are similar to 2D meshes, the $(d = 2, k = 1)$ case in Figure 5(d), where a triangular face reduces to a line. Therefore, we will mainly use this setting to describe and visualize basic concepts for simplicity. However, note that it can be generalized to any $(d, k)$ case without much problem.

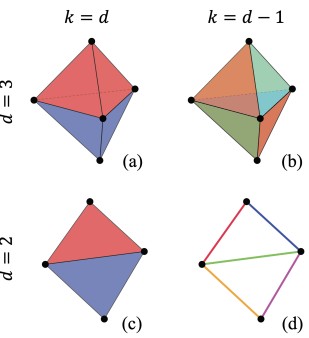

Figure 5: Renderings of $\Delta^k$s for different pairs of $(d, k)$. Different $\Delta^k$s are rendered in different colors.

---

[2]In 2D case, a vertex is a 4-dimensional vector including position (2), WDT weight (1), and real value (1).

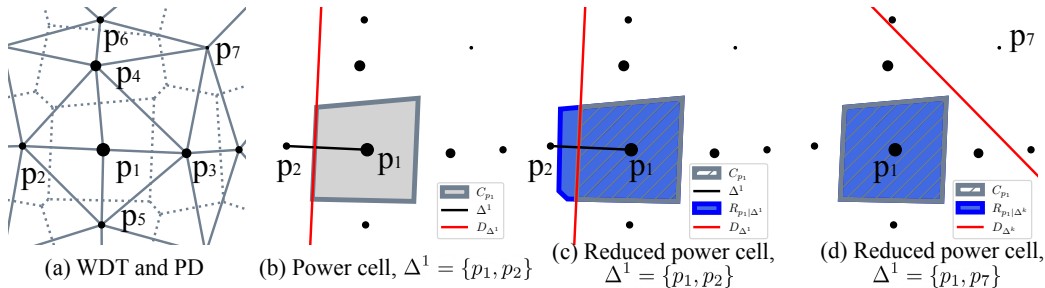

(a) WDT and PD     (b) Power cell, $\Delta^1 = \{p_1, p_2\}$     (c) Reduced power cell, $\Delta^1 = \{p_1, p_2\}$     (d) Reduced power cell, $\Delta^1 = \{p_1, p_7\}$

Figure 6: **Basic concepts to compute existence probability of given** $1$**-simplex** $(\Delta^1)$ **when** $d = 2$. **(a)**: WDT and PD of given set of weighted vertices are rendered in solid and dotted lines. The size of a vertex represents its weight. **(b)**: Power cell of $p_1$ $(C_{p_1})$ is rendered in grey. Also, $\Delta^1$ is rendered in black line, of which dual line $(D_{\Delta^1})$ is rendered in red. **(c)**, **(d)**: For given $\Delta^1$, reduced power cell of $p_1$ for the $\Delta^1$ $(R_{p_1|\Delta^1})$ is rendered in blue, with the original power cell (grey). We can evaluate the existence of $\Delta^1$ in WDT by computing the signed distance from $D_{\Delta^1}$ to $R_{p_1|\Delta^1}$.

We generalize the basic principles suggested by Rakotosaona u. a. (2021) to address our cases. For formal definitions of the concepts in this section, please refer to Appendix B.

Let $S$ be a finite set of points in $\mathbb{R}^{d=2}$ with weights. For a given point $p \in S$, we denote its weight as $w_p$. We call those weighted points in $S$ as "vertices", to distinguish them from general unweighted points in $\mathbb{R}^2$. Then, we adopt power distance $\pi(p, q) = d(p, q)^2 - w_p - w_q$ as the distance measure between two vertices in $S$. As depicted in Figure 6, $\Delta^{k=1}$ is a 1-simplex, which is a line connecting two vertices $p_i$ and $p_j$ in $S$. Its dual form $D_{\Delta^1}$ (red line) is the set of unweighted points[3] in $\mathbb{R}^2$ that are located at the same power distance to $p_i$ and $p_j$. In the power diagram, the power cell $C_p$ (gray cell) of a vertex $p$ is the set of unweighted points that are closer to $p$ than to any other vertices in $S$. We can use $C_p$ and $D_{\Delta^1}$ to measure the existence of $\Delta^1$. From Figure 6, we can see that when $\Delta^1 = \{p_i, p_j\}$ exists in WDT, its dual line $D_{\Delta^1}$ aligns exactly with $C_{p_i}$'s boundary, while when $\Delta^1 = \{p_i, p_j\}$ doesn't exist in WDT, $D_{\Delta^1}$ is outside $C_{p_i}$.

To make this measurement less binary when $\Delta^1$ exists, we use the expanded version of power cell called "reduced" power cell $(R_{p|\Delta})$, introduced by Rakotosaona u. a. (2021). The reduced power cell of $p_i \in S$ for $\Delta^1 = \{p_i, p_j\}$ is computed by excluding $p_j$ from $S$ when constructing the power cell[4]. For example, when $\Delta^1$ exists, $R_{p_i|\Delta^1}$ will expand towards $p_j$'s direction (Figure 6(c)), and $D_{\Delta^1}$ will "go through" $R_{p_i|\Delta^1}$. In contrast, when $\Delta$ doesn't exists, even though we have removed $p_j$, $R_{p_i|\Delta^1}$ will not expand (Figure 6(d)), and thus $D_{\Delta^1}$ stays outside of $R_{p_i|\Delta^1}$.

Now we can define a signed distance field $\tau_{pt}(x, R)$ for a given reduced power cell $R$, where the signed distance is measured as the distance from the point $x \in \mathbb{R}^d$ to the boundary of the reduced power cell (sign is positive when inside). Then, we can induce the signed distance between a dual form $D$ and a reduced power cell $R$:

$$\tau(D, R) = \max_{x \in D} \tau_{pt}(x, R). \tag{3}$$

As illustrate in Figure 6(c) and (d), $\tau(D_{\Delta^1}, R_{p_1|\Delta^1})$ is positive when $\Delta^1$ exists, while negative when it does not exist in WDT. This observation can be generalized as follows:

**Remark 3.1.** $\Delta^k$ exists in WDT if and only if $\forall p_i \in \Delta^k, \tau(D_{\Delta^k}, R_{p_i|\Delta^k}) > 0$.

In fact, the sign of every $\tau(D_{\Delta^k}, R_{p_i|\Delta^k})$ is same for every $p_i \in \Delta^k$. Therefore, we can use its average to measure the existence probability of $\Delta_k$, along with sigmoid function $\sigma$:

$$\Lambda_{wdt}(\Delta^k) = \frac{1}{k+1} \sum_{p \in \{\Delta^k\}} \sigma(\tau(D_{\Delta_k}, R_{p|\Delta^k}) \cdot \alpha_{wdt}), \tag{4}$$

where $\alpha_{wdt}$ is a constant value used for the sigmoid function. $\Lambda_{wdt}(\Delta^k)$ is greater than 0.5 when $\Delta^k$ exists, aligning with our probabilistic viewpoint and being differentiable.

---

[3]We treat unweighted points' weight as 0 when computing the power distance.

[4]In 3D case where $k = 2$ and $\Delta^2 = \{p_i, p_j, p_k\}$, $p_j$ and $p_k$ would be ignored for $R_{p_i|\Delta^2}$.

**Computational Challenges.** As mentioned before, Rakotosaona et al. (2021) solved the problem for the case where $(d = k = 2)$ where the dual $D_{\Delta^k}$ is a single point. Naïvely applying their approach for computing Eq. 3 to our cases poses two computational challenges:

- **Precision**: When $(d = 3, k = 2)$ or $(d = 2, k = 1)$, $D_{\Delta_k}$ becomes a **line**, not a single point. Finding a point on the line that maximizes Eq. 3 is not straightforward.
- **Efficiency**: When naïvely estimating the reduced power cell in exhaustive manner, the computational cost increases with the number of points at a rate of $O(N^2)$, where $N$ is the number of points.

See Appendix B.2 for a detailed discussion of these limitations.

## 4 Formulation

### 4.1 Practical approach to compute $\Lambda_{wdt}$

We introduce a practical approach to resolve the two aforementioned challenges. Specifically, we propose constructing the PD first and use it for computing lower bound of Eq. 3 in an efficient way. We also propose to handle the two cases separately: whether $\Delta^k$ exists in the WDT or not.

First, when the simplex $\Delta^k$ does not exist, we choose to use the negative distance between the dual form and the normal power cell, $-d(D_{\Delta^k}, C_p)$. This is the lower bound of Eq. 3:

$$-d(D_{\Delta^k}, C_p) \leq \tau(D_{\Delta^k}, R_{p|\Delta^k}) < 0, \tag{5}$$

as $C_p \subset R_{p|\Delta^k}$. See Figure 6(d) for this case in $(d = 2, k = 1)$ case. We can observe that computing this distance is computationally efficient, because the normal PD only needs to be computed once for all in advance. Moreover, $C_p$ is a convex polyhedron, and $D_{\Delta^k}$ is a (convex) line, which allows us to find the distance between line segments on the boundary of $C_p$[5] and $D_{\Delta^k}$, and choose the minimum.

Second, we analyze the case when the simplex $\Delta^k$ exists in WDT. In this case, we have to construct the reduced power cell $R_{p|\Delta^k}$ for given $\Delta^k$, which requires much additional cost. Instead of doing it, we leverage pre-computed PD to approximate the reduced power cell. Then, we pick a point $v \in D_{\Delta^k} \cap R_{p|\Delta^k}$, where $p \in \{\Delta^k\}$, and the following holds:

$$0 \leq \tau_{pt}(v, R_{p|\Delta^k}) \leq \tau(D_{\Delta^k}, R_{p|\Delta^k}), \tag{6}$$

because $v \in R_{p|\Delta^k}$ and by the definition at Eq. 3. In our case, since $D_{\Delta^k} \cap R_{p|\Delta^k}$ is a line segment, we choose its middle point as $v$ to tighten this lower bound. See Figure 6(c) for the line segment in $(d = 2, k = 1)$ case. We use this bound when $\Delta^k$ exists. Note that computing this lower bound is also computationally efficient, because we can simply project $v$ to the reduced power cell.

To sum up, we can rewrite Eq. 3 as follows.

$$\tau(D_{\Delta^k}, R_{p|\Delta^k}) = \begin{cases} \tau_{pt}(v, R_{p|\Delta^k}) & \text{if} \quad \Delta^k \in \text{WDT} \\ -d(D_{\Delta^k}, C_p) & \text{else} \end{cases} \tag{7}$$

By using this relaxation, we can get lower bound of Eq. 3, which is reliable because it always has the same sign. Also, we can reduce the computational cost from $O(N^2)$ to nearly $O(N)$, which is prerequisite for representing meshes that have more than $1K$ vertices in general. See Appendix B.2 for the computational speed and accuracy of our method, compared to the previous one. Finally, we implemented our algorithm for computing Eq. 7 in CUDA for further acceleration.

### 4.2 Definition of $\Lambda_{real}$

$\Lambda_{real}$ evaluates the existence probability of a $k$-simplex $\Delta^k$ in our mesh when it exists in WDT. To define it, we leverage per-point value $\psi \in [0, 1]$. To be specific, we compute the minimum $\psi$ of the points in $\Delta^k$ in differentiable way: $\Lambda_{real}(\Delta^k) = \sum_{p \in \Delta^k} \kappa(p) \cdot \Psi(p)$, where $\kappa(p) = e^{-\beta \cdot \Psi(p)} / \sum_{q \in \Delta^k} e^{-\beta \cdot \Psi(q)}$, and $\Psi$ is function that maps a point $p$ to its $\psi$ value. We set $\beta = 100$.

Along with $\Lambda_{wdt}$ that we discussed before, now we can evaluate the final existence probability of faces in Eq. 2. We also note here, that when we extract the final mesh, we only select the faces of which $\Lambda_{wdt}$ and $\Lambda_{real}$ are larger than 0.5.

---

[5]This holds when $d = 3$. When $d = 2$, we can use vertices of of $C_p$.

### 4.3 Loss Functions

DMesh can be reconstructed from various inputs, such as normal meshes, point clouds, and multi-view images. With its per-vertex features and per-face existence probabilities $\Lambda(F)$, we can optimize it with various reconstruction losses and regularization terms. Please see details in Appendix C.

#### 4.3.1 Reconstruction Loss ($L_{recon}$)

First, if we have a normal mesh with vertices $\mathbb{P}$ and faces $\mathbb{F}$, and we want to represent it with DMesh, we should compute the additional two per-vertex attributes, WDT weights and real values. We optimize them by maximizing $\Lambda(\mathbb{F})$ since these faces lies on the reference mesh. Conversely, for the remaining set of faces $\bar{\mathbb{F}}$ that can be defined on $\mathbb{P}$, we should minimize $\Lambda(\bar{\mathbb{F}})$. Together, they define the reconstruction loss for mesh input (Appendix C.1).

For reconstruction from point clouds or multi-view images, we need to optimize for all features including positions. For point clouds, we define our loss using Chamfer Distance (CD) and compute the expected CD using our face probabilities (Appendix C.2). For multi-view images, we define the loss as the $L_1$ loss between the given images and the rendering of DMesh, interpreting face probabilities as face opacities. We implemented efficient differentiable renderers to allow gradients to flow across face opacities (Appendix C.3).

#### 4.3.2 Regularizations

Being fully differentiable for both vertices and faces, DMesh allows us to develop various regularizations to improve the optimization process and enhance the final mesh quality. The first is **weight regularization** ($L_{weight}$), applied to the dual Power Diagram of the WDT (Appendix C.4). This regularization reduces the structural complexity of the WDT, controlling the final mesh complexity (Figure 7). The next

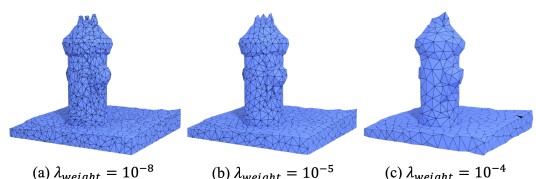

(a) $\lambda_{weight} = 10^{-8}$    (b) $\lambda_{weight} = 10^{-5}$    (c) $\lambda_{weight} = 10^{-4}$

Figure 7: **Results with different $\lambda_{weight}$.**

is **real regularization** ($L_{real}$), which enforces nearby points to have similar real values and increases the real values of points adjacent to high real value points (Appendix C.5). This helps remove holes or inner structures and makes faces near the current surface more likely to be considered (Appendix D). The final regularization, **quality regularization** ($L_{qual}$), aims to improve the quality of triangle faces by minimizing the average expected aspect ratio of the faces, thus removing thin triangles (Appendix C.6).

To sum up, our final loss function can be written as follows:

$$L = L_{recon} + \lambda_{weight} \cdot L_{weight} + \lambda_{real} \cdot L_{real} + \lambda_{qual} \cdot L_{qual},$$

where $\lambda$ values are hyperparameters. In Appendix E, we provide values for these hyperparameters for every experiment. Also, in Appendix E.3, we present ablation studies for these regularizations.

## 5 Experiments and Applications

In this section, we provide experimental results to demonstrate the efficacy of our approach. First, we optimize vertex attributes to restore a given ground truth mesh, directly proving the differentiability of our design. Next, we conduct experiments on 3D reconstruction from point clouds and multi-view images, showcasing how our differentiable formulation can be used in downstream applications.

Table 1: Mesh reconstruction results.

| -  | Bunny   | Dragon  | Buddha  |
|----|---------|---------|---------|
| RE | 99.78%  | 99.72%  | 99.64%  |
| FP | 0.00%   | 0.55%   | 0.84%   |

For the mesh reconstruction problem, we used three models from the Stanford 3D Scanning Repository (Curless und Levoy, 1996). For point cloud and multi-view reconstruction tasks, we used four closed-surface models from the Thingi32 dataset, four open-surface models from the DeepFashion3D dataset, and three additional models with both closed and open surfaces from the Objaverse dataset

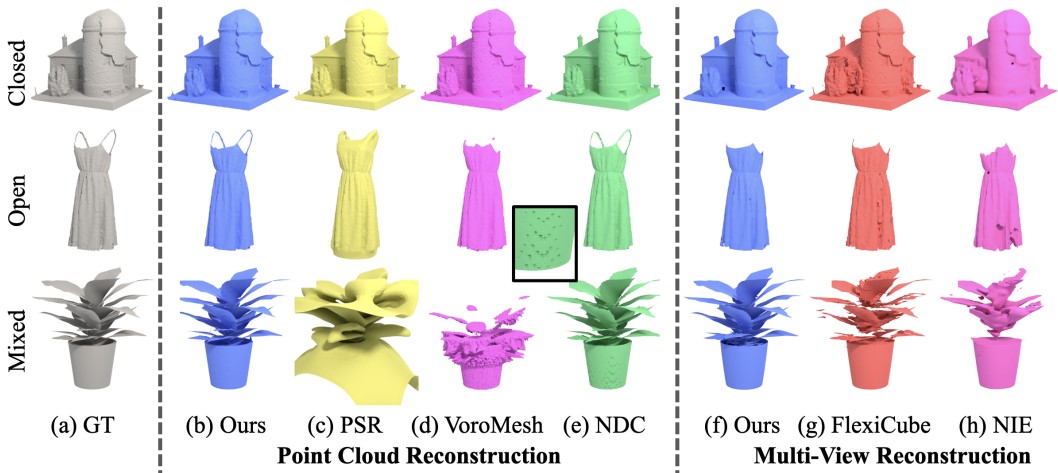

Figure 8: **Point cloud and multi-view reconstruction results. (a)**: Ground truth mesh. **(b), (f)**: Our method restores the original shape without losing much detail. **(c), (d), (g), (h)**: PSR (Kazhdan und Hoppe, 2013), VoroMesh (Maruani u. a., 2023), FlexiCube (Shen u. a., 2023), and NIE (Mehta u. a., 2022) fail for open and mixed surfaces. **(e)**: NDC (Chen u. a., 2022b) exhibits artifacts from grids.

Table 2: **Quantitative comparison for point cloud and multi-view reconstruction results.** Best results are written in bold.

|  | Methods | CD ($\times 10^{-5}$)↓ | F1↑ | NC↑ | ECD↓ | EF1↑ | # Verts↓ | # Faces↓ | Time (sec)↓ |
|---|---|---|---|---|---|---|---|---|---|
|  | PSR | 690 | 0.770 | 0.931 | 0.209 | 0.129 | 159K | 319K | 10.6 |
|  | VoroMesh | >1K | 0.671 | 0.819 | >1K | 0.263 | 121K | 242K | 12.2 |
| PC | NDC | 3.611 | 0.874 | 0.936 | **0.022** | 0.421 | 20.7K | 42.8K | **3.49** |
|  | Ours (w/o normal) | 3.726 | 0.866 | 0.936 | 0.067 | 0.342 | 3.87K | 10.4K | 775 |
|  | Ours (w/ normal) | **3.364** | **0.886** | **0.952** | 0.141 | **0.438** | **3.56K** | **7.54K** | 743 |
| MV | NIE | 585 | 0.439 | 0.848 | 0.064 | 0.023 | 74.5K | 149K | 6696 |
|  | FlexiCube | 273 | 0.591 | 0.881 | **0.039** | **0.152** | 10.9K | 21.9K | **56.47** |
|  | Ours | **34.6** | **0.685** | **0.892** | 0.094 | 0.113 | **4.19K** | **8.80K** | 1434 |

and Adobe Stock. These models are categorized as "closed," "open," and "mixed" in this section. Additionally, we use nonconvex polyhedra of various Euler characteristics and non-orientable geometries to prove our method's versatility.

We implemented our main algorithm for computing face existence probabilites and differentiable renderer used for multi-view image reconstruction in CUDA (Nickolls u. a., 2008). Since we need to compute WDT before running the CUDA algorithm, we used WDT implementation of CGAL (Jamin u. a., 2023). We implemented the rest of logic with Pytorch (Paszke u. a., 2017). All of the experiments were run on a system with AMD EPYC 7R32 CPU and Nvidia A10 GPU.

## 5.1 Mesh to DMesh

In this experiment, we demonstrate that we can preserve most of the faces in the original normal triangular mesh after converting it to DMesh using the mesh reconstruction loss introduced in 4.3.

In Table 1, we show the recovery ratio (RE) and false positive ratio (FP) of faces in our reconstructed mesh. Note that we could recover over 99% of faces in the original mesh, while only having under 1% of false faces. Please see Appendix E.1 for more details. This result successfully validates our differentiable formulation, but also reveals its limitation in reconstructing some abnormal triangles in the original mesh, such as long, thin triangles.

## 5.2 Point Cloud & Multi-View Reconstruction

In this experiment, we aim to reconstruct a mesh from partial geometric data, such as (oriented) point clouds or multi-view images. For point cloud reconstruction, we sampled 100K points from the ground truth mesh. We can additionally use point orientations, if they are available. For multi-view reconstruction, we rendered diffuse and depth images of the ground truth mesh from 64 view points.

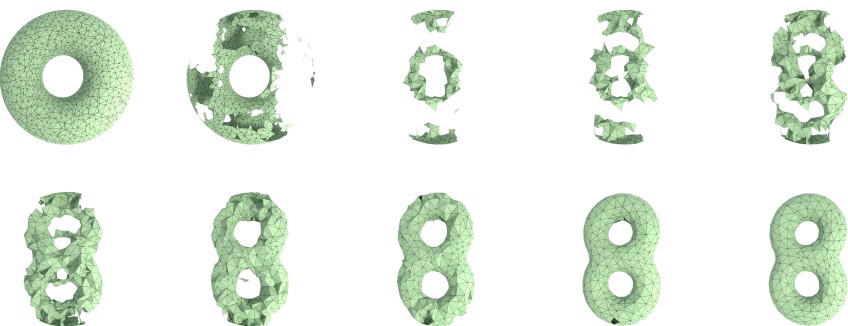

Figure 9: ($\rightarrow$) **Shape interpolation using DMesh exhibiting topology change.** After fitting DMesh to a torus (upper left), we optimize it again to reconstruct a double torus (lower right), which has a different genus. We use multi-view images for the optimization.

In Appendix E, we illustrated the example inputs for these experiments. Also, please see Appendix D to see the initialization and densification strategy we took in these experiments.

To validate our approach, we compare our results with various approaches. When it comes to point cloud reconstruction, we first compare our result with classical Screened Poisson Surface Reconstruction (PSR) method (Kazhdan und Hoppe, 2013) [6]. Then, to compare our method with optimization based approach, we use recent VoroMesh (Maruani u. a., 2023) method. Note that these two methods are not tailored for open surfaces. To compare our method also for the open surfaces, we use Neural Dual Contouring (NDC) (Chen u. a., 2022b), even though it is learning-based approach. Finally, for multi-view reconstruction task, we compare our results with Flexicube (Shen u. a., 2023) and Neural Implicit Evolution (NIE) (Mehta u. a., 2022), which correspond to volumetric approaches that can directly produce meshes of varying geometric topology for given visual inputs.

In Figure 8, we visualize the reconstruction results along with the ground truth mesh for qualitative evaluation. For closed meshes, in general, volumetric approaches like PSR, VoroMesh, and Flexicube, capture fine details better than our methods. This is mainly because we currently have limitation in the mesh resolution that we can produce with our method. NIE, which is also based on volumetric principles, generates overly smoothed reconstruction results. However, when it comes to open or mixed mesh models, which are more ubiquitous in real applications, we can observe that these methods fail, usually with false internal structures or self-intersecting faces (Appendix E.2). Since NDC leverages unsigned information, it can handle these cases without much problem as ours. However, we can observe step-like visual artifacts coming from its usage of grid in the final output, which requires post-processing. Additionally, to show the versatility of our representation, we also visualize various shapes reconstructed from oriented point clouds in Figure 2.

Table 2 presents quantitative comparisons with other methods. We used following metrics from Chen u. a. (2022b) to measure reconstruction accuracy: Chamfer Distance (CD), F-Score (F1), Normal Consistency (NC), Edge Chamfer Distance (ECD), and Edge F-Score (EF1) to the ground truth mesh. Also, we report number of vertices and faces of the reconstructed mesh to compare mesh complexity, along with computational time. All values are average over 11 models that we used. In general, our method generates mesh of comparable, or better accuracy than the other methods. However, when it comes to ECD and EF1, which evaluate the edge quality of the reconstructed mesh, our results showed some weaknesses, because our method cannot prevent non-manifold edges yet. However, our method showed superior results in terms of mesh complexity – this is partially due to the use of weight regularization. Please see Appendix E.3 to see how the regularization works through ablation studies. Likewise, our method shows promising results in producing compact and accurate mesh. However, we also note that our method requires more computational cost than the other methods in the current implementation.

Before moving on, we present an experimental result about shape interpolation using DMesh in Figure 9. We used multi-view images to reconstruct a torus first, and then optimized the DMesh again

---

[6]We provide point orientations for PSR, which is optional for our method.

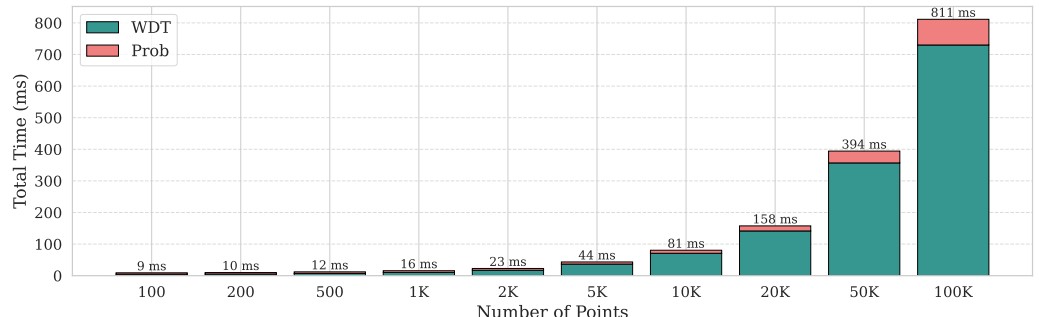

Figure 10: **Analysis of computational cost for computing face existence probabilities** ($\Lambda(F)$). The computational cost rises sharply beyond 20K points, with most of the time spent on WDT construction ("WDT"), while the probability computation ("Prob") requires significantly less time.

to fit a double torus. The results show that DMesh effectively reconstructs the double torus, even when initialized from a converged single torus, highlighting the method's robustness to local minima. However, this also indicates that our representation lacks meaningful shape interpolation, as it does not assume any specific shape topology.

# 6 Conclusion

## 6.1 Limitations

As shown above, our method achieves a more effective and complete forms of differentiable meshes of various topology than existing methods, but still has several limitations to overcome.

**Computational Cost.** Currently, the resolution of DMesh is limited by computational cost. Although our theoretical relaxation and CUDA implementation reduce this burden, processing meshes with over 100K vertices remains challenging due to the computational bottleneck of constructing the WDT at each optimization step. In Figure 10, we analyze computational costs relative to the number of points. As shown, costs rise sharply beyond 20K points, with WDT construction consuming most of the time. This limits our method's ability to handle high resolution mesh.

**Non-Manifoldness.** As we have claimed so far, DMesh shows much better generalization than the other methods as it does not have any constraints on the shape topology and mesh connectivity. However, due to this relaxation of constraint, we can observe spurious non-manifold errors in the mesh, even though we adopted measures to minimize them (Appendix D.2.7).

Specifically, an edge must have at most two adjacent faces to be a "manifold" edge. Similarly, a "manifold" vertex should be adjacent to a set of faces that form a closed or open fan. We refer to edges or vertices that do not satisfy these definitions as "non-manifold." In our results, we found that 5.50% of edges and 0.38% of vertices were non-manifold for point cloud reconstruction. For multi-view reconstruction, 6.62% of edges and 0.25% of vertices were non-manifold. Therefore, we conclude that non-manifold edges are more prevalent than non-manifold vertices in our approach.

## 6.2 Future Work

To address the computational cost issue, we can explore methods that reduce reliance on the WDT algorithm, as its cost increases significantly with the number of points. This is crucial since representing complex shapes with fine details often requires over 100K vertices. To tackle the non-manifold issue, we could integrate approaches based on (un)signed distance fields (Shen u. a., 2023; Liu u. a., 2023b) into our method, ensuring manifold mesh generation. Finally, future research could extend this work to solve other challenging problems, such as 3D reconstruction from real-world images, or applications like generative models for 3D shapes. This could involve encoding color or texture information within our framework, opening up exciting new directions for exploration.

**Acknowledgements** We thank Zhiqin Chen and Matthew Fisher for helpful advice. This research is a joint collaboration between Adobe and University of Maryland at College Park. This work has been supported in part by Adobe, IARPA, UMD-ARL Cooperate Agreement, and Dr. Barry Mersky and Capital One Endowed E-Nnovate Professorships.

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

Table 3: Traits of different optimization-based shape reconstruction methods. We compare methods based on template mesh (Palfinger, 2022; Nicolet u. a., 2021), neural SDF (Wang u. a., 2021, 2023), neural UDF (Long u. a., 2023; Liu u. a., 2023a), differentiable isosurface extraction techniques (Shen u. a., 2021; Munkberg u. a., 2022; Shen u. a., 2023; Liu u. a., 2023b) with ours.

| Methods | Closed | Open | Diff. Mesh | Diff. Render. | Geo. Topo. | Mesh Topo. | Manifold |
|---|---|---|---|---|---|---|---|
| Template Mesh | O | O | O | O | X | X | O |
| Neural SDF | O | X | X | O | O | X | O |
| Neural UDF | O | O | X | O | O | X | △ |
| Diff. Isosurface | O | △ | O | O | O | X | O |
| **DMesh (Ours)** | O | O | O | O | O | O | X |

# A  Comparison to Other Shape Reconstruction Methods

Here we provide conceptual comparisons between our approach and the other optimization-based 3D reconstruction algorithms, which use different shape representations. To be specific, we compared our method with mesh optimization methods starting from template mesh (Palfinger, 2022; Nicolet u. a., 2021), methods based on neural signed distance fields (SDF) (Wang u. a., 2021, 2023), methods based on neural unsigned distance fields (UDF) (Liu u. a., 2023a; Long u. a., 2023), and methods based on differentiable isosurface extraction (Shen u. a., 2021; Munkberg u. a., 2022; Shen u. a., 2023; Liu u. a., 2023b). We used following criteria to compare these methods.

- Closed surface: Whether or not the given method can reconstruct, or represent closed surfaces.

- Open surface: Whether or not the given method can reconstruct, or represent open surfaces.

- Differentiable Meshing: Whether or not the given method can produce gradients from the loss computed on the final mesh.

- Differentiable Rendering: Whether or not the given method can produce gradients from the loss computed on the rendering results.

- Geometric topology: Whether or not the given method can change geometric topology of the shape. Here, geometric topology defines the continuous deformation of Euclidean subspaces (Lee, 2010). For instance, genus of the shape is one of the traits that describe geometric topology.

- Mesh topology: Whether or note the given method can produce gradients from the loss computed on the mesh topology, which denotes the structural configuration, or edge connectivity of a mesh.

- Manifoldness: Whether or not the given method guarantees manifold mesh.

In Table 3, we present a comparative analysis of different methods. Note that our method meets all criteria, only except manifoldness. It is partially because our method does not assume volume, which is the same for methods based on neural UDF. However, because our method does not leverage smoothness prior of neural network like those methods, it could exhibit high frequency noises in the final mesh. Because of this reason, we gave △ to the neural UDF methods, while giving X to our approach. When it comes to methods based on differentiable isosurface extraction algorithms, we gave △ to its ability to handle open surfaces, because of (Liu u. a., 2023b). They can represent open surfaces as subset of closed ones, but cannot handle non-orientable open surfaces. Finally, note that our method is currently the only method that can handle mesh topology.

Likewise, DMesh shows promise in addressing the shortcomings found in previous research. Nonetheless, it has its own set of limitations (Section 6). Identifying and addressing these limitations is crucial for unlocking the full potential of our method.

# B  Details about Section 3.2

## B.1  Mathematical Definitions

Here, we provide formal mathematical definitions of the terms used in Section 3.2. We mainly use notations from Aurenhammer u. a. (2013) and Cheng u. a. (2013).

Generalizing the notations in Section 3.2, let $S[w]$ be a finite set of weighted points in $\mathbb{R}^d$, where $w$ is a weight assignment that maps each point $p \in S$ to its weight $w_p$. We denote a weighted point as $p[w_p]$ and define power distance to measure distance between two weighted points.

**Definition B.1** (Power distance). Power distance between two weighted points $p[w_p]$ and $q[w_q]$ is measured as:

$$\pi(p[w_p], q[w_q]) = d(p, q)^2 - w_p - w_q, \tag{8}$$

where $d(p, q)$ is the Euclidean distance.

Note that an unweighted point is regarded as carrying weight of 0. Based on this power distance, we can define the power cell $C_p$ of a point $p[w_p]$ as the set of unweighted points in $\mathbb{R}^d$ that are closer to $p[w_p]$ than to any other weighted point in $S[w]$.

**Definition B.2** (Power cell). Power cell of a point $p[w_p] \in S[w]$ is defined as:

$$C_p = \{x \in \mathbb{R}^d \mid \forall q[w_q] \in S[w], \pi(x, p[w_p]) \le \pi(x, q[w_q])\}. \tag{9}$$

Note that some points may have empty power cells if their weights are relatively smaller than neighboring points. We call them "submerged" points. As we will see later, weight regularization aims at submerging unnecessary points in mesh definition, which leads to mesh simplification.

To construct the power cell, we can use the concept of half space. A half space $H_{p<q}$ is the set of unweighted points in $\mathbb{R}^d$ that are closer to $p[w_p]$ than $q[w_q]$.

**Definition B.3** (Half space). Half space $H_{p<q}$ is defined as:

$$H_{p<q} = \{x \in \mathbb{R}^d \mid \pi(x, p[w_p]) \le \pi(x, q[w_q])\}. \tag{10}$$

Note that we can construct a power cell by intersecting half spaces, which proves the convexity of the power cell. Now we call $H(p, q)$ as a half plane that divides $\mathbb{R}^d$ into two half spaces, $H_{p<q}$ and $H_{q<p}$.

**Definition B.4** (Half plane). Half plane $H_{p,q}$ is defined as:

$$H_{p,q} = \{x \in \mathbb{R}^d \mid \pi(x, p[w_p]) = \pi(x, q[w_q])\}. \tag{11}$$

Then, for a given $k$-simplex $\Delta^k$ comprised of weighted points $\{\Delta^k\} = \{p_1[w_{p_1}], \ldots, p_{k+1}[w_{p_{k+1}}]\} \subset S[w]$, the dual structure $D_{\Delta^k}$ is the intersection of half planes between the points in $\{\Delta^k\}$, which is a convex set.

**Definition B.5** (Dual form). Dual form $D_{\Delta^k}$ of $\Delta^k = \{p_1[w_{p_1}], \ldots, p_{k+1}[w_{p_{k+1}}]\}$ is defined as:

$$D_{\Delta^k} = \bigcap_{i,j=1,\ldots,k+1} H(p_i, p_j), \tag{12}$$

which is equivalent to:

$$D_{\Delta^k} = \{x \in \mathbb{R}^d \mid \forall p_i[w_{p_i}], p_j[w_{p_j}] \in S[w], \pi(x, p_i[w_{p_i}]) = \pi(x, p_j[w_{p_j}])\}. \tag{13}$$

Note than when $k = d$, $D_{\Delta^k}$ becomes a point, while it becomes a line when $k = d - 1$. As discussed in Section 3.2, we leverage the distance between this dual form $D_{\Delta^k}$ and reduced power cell (Rakotosaona u. a., 2021) of the points in $D_{\Delta^k}$ to query the existence of $\Delta^k$. The reduced power cell $R_{p|\Delta^k}$ is a power cell of $p$ that does not concern the other points in $\Delta^k$ in its construction.

**Definition B.6** (Reduced power cell). Reduced power cell of a weighted point $p[w] \in S[w]$ for given $\Delta^k$ is defined as:

$$R_{p|\Delta^k} = \{x \in \mathbb{R}^d \mid \forall q[w_q] \in S[w] - \{\Delta^k\}, \pi(x, p[w_p]) \le \pi(x, q[w_q])\}. \tag{14}$$

Using these concepts, we can measure the existence probability of a $k$-simplex, as provided in Section 3.2.

## B.2 Analysis of previous approach

### B.2.1 Theoretical Aspect

As mentioned in Section 3.2, the previous approach of Rakotosaona u. a. (2021) has two computational challenges in computational efficiency and precision. These limitations are mainly rooted in not knowing the power diagram structure before evaluating Eq. 3. We summarize the overall procedure of the previous approach, and point out how our method is different from it.

**Collecting simplexes to evaluate**    First, we need to collect simplexes that we want to compute probabilities for. Here we assume the number of simplexes increases linearly ($O(N)$) as the number of points ($N$) increases. This is a plausible assumption, because we often search for $k$-nearest neighbors for each point, and combine them to generate the query simplexes.

**Sampling a point on the dual forms of the simplexes**    The previous method relies on point projections to evaluate Eq. 3. This did not incur any problem for their case ($d = k = 2$), because the dual form was a single point. However, when ($k < d$) as in our cases, the dual form contains infinite number of points, which makes unclear how to apply this point-based approach. One possible solution is sampling the most "representative" point on the dual form, and leveraging the point to estimate Eq. 3. By definition, this estimation is lower bound of Eq. 3. However, the problem arises when this sample point does not give reliable result. For instance, we can consider the case shown in Figure 6(c). In the illustration, we can observe that $\Delta^1$ exists in WDT, and thus $D_{\Delta^1}$ goes thorugh $R_{p_1|D_{\Delta^1}}$. If we sample a point on $D_{\Delta^1}$ that is included in $R_{p_1|D_{\Delta^1}}$, the signed distance from the sample point would have same sign as Eq. 3. However, if the sample point is selected outside $R_{p_1|D_{\Delta^1}}$, the sign would be different from the real value. In this case, even if $\Delta^k$ exists, we can recognize it as not existing. Note that this false estimation produces false gradients, which could undermine optimization process.

In contrast, we do not have to concern about this precision issue, because we construct PD, which tells us good sample points that give reliable lower bounds of Eq. 3, when the given simplex exists in WDT. Otherwise, we explicitly compute minimum distance between the dual form and the reduced power cell, as discussed in Section 4.1.

**Projecting sample points to reduced power cells**    The final step is point projection, where we project the sample points from dual forms to the reduced power cells to estimate Eq. 3. Based on the definitions in Appendix B, we can observe that a (reduced) power cell's boundaries are comprised of half planes. That is, the boundaries of $C_p$ is comprised of multiple half planes between $p$ and the other weighted points. However, when we do not know which half planes comprise the boundaries, we have to do exhaustive search to find the signed distance from the sample point to the boundaries of the reduced power cell. As the number of half planes that are associated with a point $p$ is $N$, the computational cost to precisely compute the signed distance is $O(N)$.

Note that it does not hold for our case, because by constructing WDT and PD, we know which half planes form the boundaries of each power cell. Also, note that even when the number of points increases, the average number of half planes that comprise the boundaries of power cells remains constant. Therefore, in our case, this step requires only $O(1)$ computational cost.

**Summary**    To sum up, the computational cost of the previous approach amounts to $O(N^2)$, as the number of simplexes to evaluate increases linearly, and the cost for the projection step also increases linearly as the number of points increase. However, the cost for ours remains at $O(N)$. Moreover, the previous approach does not guarantee satisfactory estimations of Eq. 3.

Before moving on, we point out that the original implementation limited the number of half planes to consider in evaluating Eq. 3 to reduce the computational cost to $O(N)$. This relaxation is permissible to the case where the precision is not very important. However, the precision is important in our case, because we aim at representing mesh accurately.

### B.2.2 Experimental Results

To prove the aforementioned theoretical claim, we conducted experiments to measure the computational speed and accuracy of the probability estimation for ($d = 2, k = 2$) and ($d = 2, k = 1$) cases, for varying number of points. We randomly sampled points in a unit cube uniformly, and set the

Table 4: Comparison of computational speed and accuracy between the previous method (Rakotosaona u. a., 2021) and ours. For each number of points in 2D, we report the triplet of (computational speed (ms) / false positive ratio (%) / false negative ratio (%)), along with the number of query simplexes that we give to the algorithm.

| | # Pts. | 100 | 300 | 1K | 3K | 10K | 30K |
|---|---|---|---|---|---|---|---|
| | Prev. | 6.38 / 0 / 0 | 19.7 / 0 / 0 | 139 / 0 / 0 | 739 / 0 / 0.16 | 4376 / 0 / 0.12 | - |
| (d=2, k=2) | Ours. | 105 / 0 / 0 | 107 / 0 / 0 | 108 / 0 / 0 | 111 / 0 / 0 | 116 / 0 / 0.12 | 125 / 0 / 0.25 |
| | # Simp. | 342 | 992 | 2274 | 3672 | 4974 | 6332 |
| | Prev. | 5.95 / 0 / 49.9 | 16.9 / 0 / 49.8 | 116 / 0 / 49.9 | 609 / 0 / 49.9 | 3681 / 0 / 49.9 | - |
| (d=2, k=1) | Ours. | 99.4 / 0 / 0 | 100 / 0 / 0 | 103 / 0 / 0 | 105 / 0.03 / 0 | 111 / 0 / 0.08 | 123 / 0 / 0.29 |
| | # Simp. | 526 | 1504 | 3428 | 5522 | 7482 | 9524 |

weights by random sampling from a normal distribution: $\mathcal{N}(0, 10^{-3})$. For fair comparison, we did not use CUDA implementation that we mainly use in the paper. We implemented both algorithms in PyTorch, and ran 10 times for each setting to get fair values. In each experiment, we fed the query simplexes into the algorithm, and computed their existence probabilities. If the computed probability for a simplex was over 0.5, but did not exist, it is counted to false positive ratio. In contrast, if the computed probability for a simplex was below 0.5, but did exist, it is counted to false negative ratio. We measured the computational accuracy with these metrics.

In Table 4, we can see that as the number of points increases, the computational cost increases exponentially for the previous approach, while ours remain fairly stable up to 30K points. When the number of points reached 100K, the previous method failed because of excessive memory consumption. However, when the number of points is smaller than 1K, the previous method ran faster than ours, because we need to construct PD even for the small number of points, which consumes most of the time for all these cases.

In terms of accuracy, we can observe that the previous method and ours both give accurate estimations when $(d = 2, k = 1)$. However, when $(d = 2, k = 1)$, we can see that the false negative ratio is almost $50\%$ – as we set the number of existing simplexes and non-existing simplexes in the query set as the same, it means that the previous method chose wrong sample points for most of the query simplexes, so that it predicted most of them as not existing. To select the most representative point, we found the intersection point between the dual form and the affine space created by the weighted points in the simplex. This could be another interesting direction to explore, but this approach found out to be not very accurate in this experiment. In contrast, our approach gave stable estimations for all the cases. This result demonstrates that our method is more scalable, and gives reliable probability estimations than the previous approach, which is necessary for accurate 3D mesh representation.

## C  Loss Functions

Here we provide formal definitions for the loss functions that we use in the paper.

### C.1  Mesh to DMesh

In this section, we explore the loss function used to transform the ground truth mesh into our DMesh representation. As previously mentioned in Section 4.3, the explicit definition of ground truth connectivity in the provided mesh allows us to establish a loss function based on it.

Building on the explanation in Section 4.3, if the ground truth mesh consists of vertices $\mathbb{P}$ and faces $\mathbb{F}$, we can construct an additional set of faces $\bar{\mathbb{F}}$. These faces are formed from vertices in $\mathbb{P}$ but do not intersect with faces in $\mathbb{F}$.

$$\bar{\mathbb{F}} = \mathbb{F}^* - \mathbb{F}, \text{ where } \mathbb{F}^* = \text{ every possible face combination on } \mathbb{P}.$$

Then, we notice that we should maximize the existence probabilities of faces in $\mathbb{F}$, but minimize those of faces in $\bar{\mathbb{F}}$. Therefore, we can define our reconstruction loss function as

$$L_{recon} = - \sum_{F \in \mathbb{F}} \Lambda(F) + \sum_{F \in \bar{\mathbb{F}}} \Lambda(F). \tag{15}$$

If the first term of the loss function mentioned above is not fully optimized, it could lead to the omission of ground truth faces, resulting in a poorer recovery ratio (Section 5.1). Conversely, if the

second term is not fully optimized, the resulting DMesh might include faces absent in the ground truth mesh, leading to a higher false positive ratio (Section 5.1). Refer to Appendix D.1 for details on how this reconstruction loss is integrated into the overall optimization process.

## C.2 Point Cloud Reconstruction

In the task of point cloud reconstruction, we reconstruct the mesh by minimizing the ($L_1$-norm based) expected Chamfer Distance (CD) between the given point cloud ($\mathbb{P}_{gt}$) and the sample points ($\mathbb{P}_{ours}$) from our reconstructed mesh. We denote the CD from $\mathbb{P}_{gt}$ to $\mathbb{P}_{ours}$ as $CD_{gt}$, and the CD from $\mathbb{P}_{ours}$ to $\mathbb{P}_{gt}$ as $CD_{ours}$. The final reconstruction loss is obtained by combining these two distances.

$$L_{recon} = CD_{gt} + CD_{ours}. \tag{16}$$

### C.2.1 Sampling $\mathbb{P}_{ours}$

To compute these terms, we start by sampling $\mathbb{P}_{ours}$ from our current mesh. First, we sample a set of faces that we will sample points from. We consider the areas of the triangular faces and their existence probabilities. To be specific, we define $\eta(F)$ for a face $F$ as

$$\bar{\eta}(F) = \Lambda(F), \quad \eta(F) = F_{area} \cdot \bar{\eta}(F),$$

and define a probability to sample $F$ from the entires faces $\mathbb{F}$ as

$$P_{sample}(F) = \frac{\eta(F)}{\sum_{F' \in \mathbb{F}} \eta(F')}.$$

We sample $N$ faces from $\mathbb{F}$ with replacement and then uniformly sample a single point from each selected face to define $\mathbb{P}_{ours}$. In our experiments, we set $N$ to $100K$.

In this formulation, we sample more points from faces with a larger area and higher existence probability to improve sampling efficiency. However, we observed that despite these measures, the sampling efficiency remains low, leading to slow convergence. This issue arises because, during optimization, there is an excessive number of faces with very low existence probability.

To overcome this limitation, we decided to do stratified sampling based on point-wise real values and cull out faces with very low existence probabilities. To be specific, we define two different $\eta$ functions:

$$\bar{\eta}_1(F) = \Lambda_{wdt}(F) \cdot \min(\psi_i, \psi_j, \psi_k), \quad \eta_1(F) = F_{area} \cdot \bar{\eta}_1(F)$$
$$\bar{\eta}_2(F) = \Lambda_{wdt}(F) \cdot \max(\psi_i, \psi_j, \psi_k), \quad \eta_2(F) = F_{area} \cdot \bar{\eta}_2(F)$$

where $(\psi_i, \psi_j, \psi_k)$ are the real values of the points that comprise $F$. Note that $\eta_1$ is the same as $\eta$ [7].

For the faces in $\mathbb{F}$, we first calculate the $\bar{\eta}_1$ and $\bar{\eta}_2$ values and eliminate faces with values lower than a predefined threshold $\epsilon_\eta$. We denote the set of remaining faces as $\mathbb{F}_1$ and $\mathbb{F}_2$. Subsequently, we sample $\frac{N}{2}$ faces from $\mathbb{F}_1$ and the other $\frac{N}{2}$ faces from $\mathbb{F}_2$, using the following two sampling probabilities:

$$P_{sample,1}(F) = \frac{\eta_1(F)}{\sum_{F' \in \mathbb{F}_1} \eta_1(F')}, \quad P_{sample,2}(F) = \frac{\eta_2(F)}{\sum_{F' \in \mathbb{F}_2} \eta_2(F')}.$$

The rationale behind this sampling strategy is to prioritize (non-existing) faces closer to the current mesh over those further away. In the original $\eta = \eta_1$ function, we focus solely on the minimum real value, leading to a higher sampling rate for existing faces. However, to remove holes in the current mesh, it's beneficial to sample more points from potential faces—those not yet existing but connected to existing ones. This approach, using $\eta_2$, enhances reconstruction results by removing holes more effectively. Yet, there's substantial potential to refine this importance sampling technique, as we haven't conducted a theoretical analysis in this study.

Moreover, when sampling a point from a face, we record the face's existence probability alongside the point. Additionally, if necessary, we obtain and store the face's normal. For a point $\mathbf{p} \in \mathbb{P}_{ours}$,

---

[7] We do not use differentiable min operator, as we do not require differentiability in the sampling process.

we introduce functions $\Lambda_{pt}(\cdot)$ and $Normal(\cdot)$ to retrieve the face existence probability and normal, respectively:

$$\Lambda_{pt}(\mathbf{p}) = \Lambda(F(\mathbf{p})), \quad Normal(\mathbf{p}) = F(\mathbf{p})_{normal},$$
$$F(\mathbf{p}) = \text{the face where } \mathbf{p} \text{ was sampled from.}$$

### C.2.2  $CD_{gt}$

Now we introduce how we compute the $CD_{gt}$, which is CD from $\mathbb{P}_{gt}$ to $\mathbb{P}_{ours}$. For each point $\mathbf{p} \in \mathbb{P}_{gt}$, we first find $k$-nearest neighbors of $\mathbf{p}$ in $\mathbb{P}_{ours}$, which we denote as $(p_1, p_2, ..., p_k)$. Then, we define a distance function between the point $\mathbf{p}$ and the $k$-nearest neighbors as follows, to accommodate the orientation information:

$$\bar{D}(\mathbf{p}, p_i) = ||\mathbf{p} - p_i||_2 + \lambda_{normal} \cdot \bar{D}_n(\mathbf{p}, p_i),$$
$$\text{where } \bar{D}_n(\mathbf{p}, p_i) = 1 - | < \mathbf{p}_{normal}, Normal(p_i) > |, \tag{17}$$

where $\lambda_{normal}$ is a parameter than determines the importance of point orientation in reconstruction. If $\lambda_{normal} = 0$, we only consider the positional information of the sampled points.

After we evaluate the above distance function values for the $k$-nearest points, we reorder them in ascending order. Then, we compute the following expected minimum distance from $\mathbf{p}$ to $\mathbb{P}_{ours}$,

$$D(\mathbf{p}, \mathbb{P}_{ours}) = \sum_{i=1,...,k} \bar{D}(\mathbf{p}, p_i) \cdot P(p_i) \cdot \bar{P}(p_i),$$
$$P(p_i) = \Lambda_{pt}(p_i) \cdot \mathbb{I}_{prev}(F(p_i),$$
$$\bar{P}(p_i) = \Pi_{i=1,...,k-1}(1 - P(p_i)),$$

where $\mathbb{I}_{prev}$ is an indicator function that returns 1 only when the given face has not appeared before in computing the above expected distance. For instance, if the face ids for the reordered points were $(1, 2, 3, 2, 3, 4)$, the $\mathbb{I}_{prev}$ function evaluates to $(1, 1, 1, 0, 0, 1)$. This indicator function is needed, because if we select $p_i$ as the nearest point to $\mathbf{p}$ with the probability $\Lambda_{pt}(\mathbf{p})$, it means that we interpret that the face corresponding to $p_i$ already exists, and then we would select $p_i$ on the face as the nearest point to $\mathbf{p}$ rather than the other points that were sampled from the same face, but have larger distance than $p_i$ and thus come after $p_i$ in the ordered points.

Note that we dynamically change $k$ during runtime to get a reliable estimation of $D(\mathbf{p}, \mathbb{P}_{ours})$. That is, for current $k$, if most of $\bar{P}(p_k)$s for the points in $\mathbb{P}_{gt}$ are still large, it means that there is a chance that the estimation could change a lot if we find and consider more neighboring points. Therefore, in our experiments, if any point in $\mathbb{P}_{gt}$ has $\bar{P}(p_k)$ larger than $10^{-4}$, we increase $k$ by 1 for the next iteration. However, if there is no such point, we decrease $k$ by 1 to accelerate the optimization process.

Finally, we can compute $CD_{gt}$ by summing up the point-wise expected minimum distances.

$$CD_{gt} = \sum_{\mathbf{p} \in \mathbb{P}_{gt}} D(\mathbf{p}, \mathbb{P}_{ours}).$$

### C.2.3  $CD_{ours}$

In computing $CD_{ours}$, which is $CD$ from $\mathbb{P}_{ours}$ to $\mathbb{P}_{gt}$, we also find $k$-nearest neighbors for each point $\mathbf{p} \in \mathbb{P}_{ours}$, which we denote as $(p_1, p_2, ..., p_k)$. Then, for a point $\mathbf{p}$, we use the same distance function $\bar{D}$ in Eq. 17 to find the distance between $\mathbf{p}$ and $(p_1, p_2, ..., p_k)$. After that, we select the minimum one for each point, multiply the existence probability of each point, and then sum them up to compute $CD_{ours}$.

$$D(\mathbf{p}, \mathbb{P}_{gt}) = \min_{i=1,...,k} \bar{D}(\mathbf{p}, p_i),$$
$$CD_{ours} = \sum_{\mathbf{p} \in \mathbb{P}_{ours}} \Lambda_{pt}(\mathbf{p}) \cdot D(\mathbf{p}, \mathbb{P}_{gt}).$$

Finally, we can compute the final reconstruction loss for point clouds as shown in Eq. 16.

## C.3 Multi-View Reconstruction

When we are given multi-view images, we reconstruct the mesh by minimizing the $L_1$ difference between our rendered images and the given images. In this work, we mainly use both diffuse and depth renderings to reconstruct the mesh.

If we denote the ($N_{img}$) ground truth images of $N_{pixel}$ number of pixels as $\mathcal{I}_i^{gt}$ ($i = 1, ..., N_{img}$), and our rendered images as $\mathcal{I}_i^{ours}$, we can write the reconstruction loss function as

$$L_{recon} = \frac{1}{N_{img} \cdot N_{pixel}} \sum_{i=1,...,N_{img}} ||\mathcal{I}_i^{gt} - \mathcal{I}_i^{ours}||.$$

Then, we can define our rendered image as follows:

$$I_i^{ours} = \mathcal{F}(\mathbb{P}, \mathbb{F}, \Lambda(\mathbb{F}), \mathbf{MV_i}, \mathbf{P_i}).$$

where $\mathcal{F}$ is a differentiable renderer that renders the scene for the given points $\mathbb{P}$, faces $\mathbb{F}$, face existence probabilities $\Lambda(\mathbb{F})$, $i$-th modelview matrix $\mathbf{MV_i} \in \mathbb{R}^{4 \times 4}$, and $i$-th projection matrix $\mathbf{P_i} \in \mathbb{R}^{4 \times 4}$. The differentiable renderer $\mathcal{F}$ has to backpropagate gradients along $\mathbb{P}$, $\mathbb{F}$, and $\Lambda(\mathbb{F})$ to update our point attributes. Specifically, here we interpret $\Lambda(\mathbb{F})$ as opacity for faces to use in the rendering process. This is because opacity means the probability that a ray stops when it hits the face, which aligns with our face existence probability well. For this reason, we ignore faces with very low existence probability under some threshold to accelerate the reconstruction, as they are almost transparent and do not contribute to the rendering a lot.

To implement $\mathcal{F}$, we looked through previous works dedicated for differentiable rendering (Laine u. a., 2020; Liu u. a., 2019). However, we discovered that these methods incur substantial computational costs when rendering a large number of (potentially) semi-transparent triangles, as is the case in our scenario. Consequently, we developed two efficient, partially differentiable renderers that meet our specific requirements. These renderers fulfill distinct roles within our pipeline—as detailed in Appendix D, our optimization process encompasses two phases within a single epoch. The first renderer is employed during the initial phase, while the second renderer is utilized in the subsequent phase.

### C.3.1 $\mathcal{F}_A$

If there are multiple semi-transparent faces in the scene, we have to sort the faces that covers a target pixel with their (view-space) depth values, and iterate through them until the accumulated transmittance is saturated to determine the color for the pixel. Conducting this process for each individual pixel is not only costly, but also requires a lot of memory to store information for backward pass.

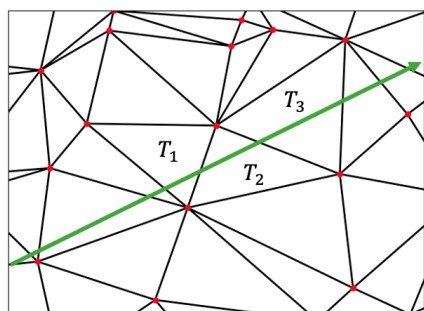

Recently, 3D Gaussian Splatting (Kerbl u. a., 2023) overcame this issue with tile-based rasterizer. We adopted this approach, and modified their implementation to render triangular faces, instead of gaussian splats. To briefly introduce its pipeline, it first assigns face-wise depth value by computing the view-space depth of its center point. Then, after subdividing the entire screen into $16 \times 16$ tiles, we assign faces to each tiles if they overlap. After that, by using the combination of tile ID and the face-wise depth as a key, we get the face list sorted by depth value in each

Figure 11: $\mathcal{F}_B$ uses tessellation structure to efficiently render overlapped faces in the correct order.

tile. Finally, for each tile, we iterate through the sorted faces and determine color and depth for each pixel as follows.

$$C = \sum_{i=1,...,k} T_i \cdot \alpha_i \cdot C_i, \quad (T_i = \Pi_{j=1,...,i-1}(1 - \alpha_j)),$$

where $T_i$ is the accumulated transmittance, $\alpha_i$ is the opacity of the $i$-th face, and $C_i$ is the color (or depth) of the $i$-th face. Note that $\alpha_i = \Lambda(F_i)$, as mentioned above.

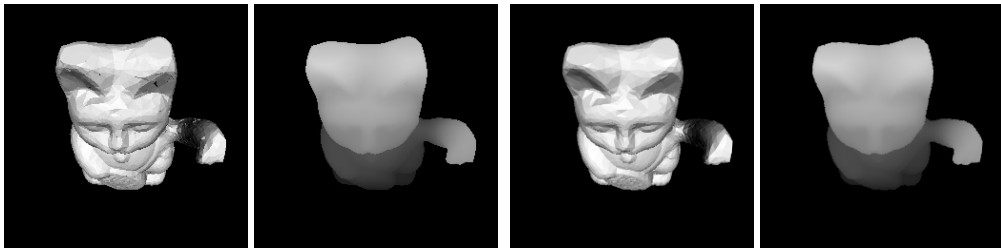

(a) Rendered Images from $\mathcal{F}_A$          (b) Rendered Images from $\mathcal{F}_{A'}$

Figure 12: Rendered images from two differentiable renderers, $\mathcal{F}_A$ and $\mathcal{F}_{A'}$. Left and right image corresponds to diffuse and depth rendering, respectively. (a) $\mathcal{F}_A$ is our (partially) differentiable renderer based on tile-based approach. (b) Since $\mathcal{F}_A$ does not produce visibility-related gradients, we additionally use $\mathcal{F}_{A'}$ (Laine u. a., 2020) to render images and integrate with ours.

Even though this renderer admits an efficient rendering of large number of semi-transparent faces, there are still two large limitations in the current implementation. First, the current implementation does not produce visibility-related gradients (near face edges) to update point attributes. Therefore, we argue that this renderer is partially differentiable, rather than fully differentiable. Next, since it does not compute precise view-point depth for each pixel, its rendering result can be misleading for some cases, as pointed out in (Kerbl u. a., 2023).

To amend the first issue, we opt to use another differentiable renderer of Laine u. a. (2020), which produces the visibility-related gradients that we lack. Since this renderer cannot render (large number of) transparent faces as ours does, we only render the faces with opacity larger than $0.5$. Also, we set the faces to be fully opaque. If we call this renderer as $\mathcal{F}_{A'}$, our final rendered image can be written as follows.

$$\mathcal{I}_i^{ours} = \frac{1}{2}(\mathcal{F}_A(\mathbb{P}, \mathbb{F}, \Lambda(\mathbb{F}), \mathbf{MV}_i, \mathbf{P}_i) + \mathcal{F}_{A'}(\mathbb{P}, \mathbb{F}, \Lambda(\mathbb{F}), \mathbf{MV}_i, \mathbf{P}_i)).$$

In Figure 12, we illustrate rendered images from $\mathcal{F}_A$ and $\mathcal{F}_{A'}$.

Acknowledging that this formulation is not theoretically correct, we believe that it is an intriguing future work to implement a fully differentiable renderer that works for our case. However, we empirically found out that we can reconstruct a wide variety of meshes with current formulation without much difficulty.

As mentioned before, this renderer is used at the first phase of the optimization process, where all of the point attributes are updated. However, in the second phase, we fix the point positions and weights, and only update point-wise real values (Appendix D.2). In this case, we can leverage the tessellation structure to implement an efficient differentiable renderer. As the second renderer does a precise depth testing unlike the first one, it can be used to modify the errors incurred by the second limitation of the first renderer (Figure 13).

### C.3.2   $\mathcal{F}_B$

The second renderer performs precise depth ordering in an efficient way, based on the fixed tessellation structure that we have. In Figure 11, we illustrate a 2D diagram that explains our approach. When the green ray, which corresponds to a single ray to determine the color of a single pixel, goes through the tessellation, we can observe that it goes through a sequence of triangles (tetrahedron in 3D), which are denoted as $T_1, T_2$, and $T_3$. When the ray enters a triangle $T_i$ through one of its three edges, we can see that it moves onto the other adjacent triangle $T_{i+1}$ only through one of the other edges of $T_i$, because of compact tessellation. Therefore, when the ray hits one edge of $T_i$, it can only examine the other two edges of $T_i$ to find the next edge it hits. Note that we do not have to do depth testing explicitly in this approach. Also, unlike the first approach, this renderer does not have to store all the possible faces that a ray collides for the backward pass, because it can iterate the same process in the opposite way in the backward pass to find the edge that it hit before the last edge. If we only store the last edge that each hits at the forward pass, we can start from the last edge and find the previous edges that it hit to compute gradients. Therefore, this second renderer requires much less memory

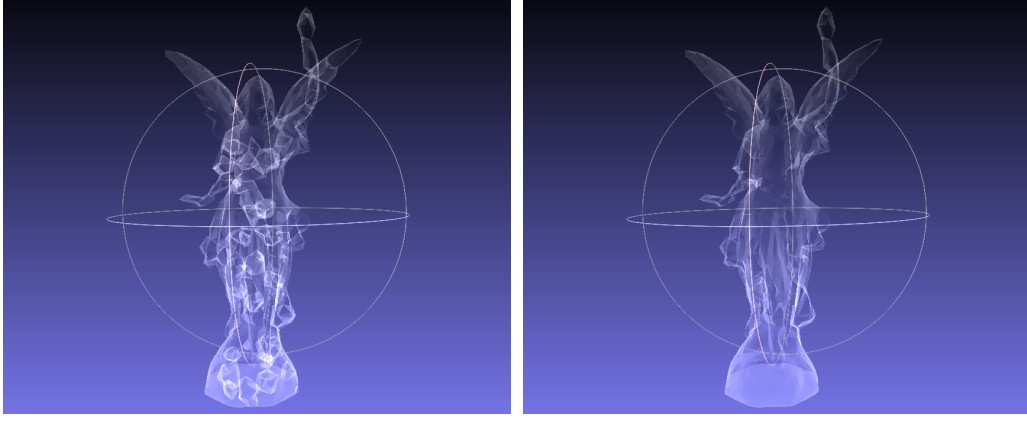

(a) Extracted Mesh after phase 1        (b) Extracted Mesh after phase 2

Figure 13: Reconstructed mesh from multi-view images, rendered in MeshLab's (Cignoni u. a., 2008) x-ray mode to see inner structure. In multi-view reconstruction, we divide each epoch in two phases. (a) After the first phase ends, where we do inaccurate depth testing, lots of false inner faces are created. (b) To remove these inner faces, we require a renderer that does the exact depth testing, which we use in the second phase. Also see Appendix D.2 for details about post-processing step to remove the inner structure.

than the first one, and also performs precise depth testing naturally. However, note that this renderer is also partilly differentiable, because it cannot update point positions and weights.

To sum up, we implemented two partially differentiable renderers to solve multi-view reconstruction problem with DMesh. They serve different objectives in our reconstruction process, and we empirically found out that they are powerful enough to reconstruct target meshes in our experiments. However, we expect that we can simplify the process and improve its stability, if we can implement a fully differentiable renderer that satisfy our needs. We leave it as a future work.

### C.4 Weight Regularization

Weight regularization aims at reducing the complexity of WDT, which supports our mesh. By using this regularization, we can discard unnecessary points that do not contribute to representing our mesh. Moreover, we can reduce the number of points on the mesh, if they are redundant, which ends up in the mesh simplification effect (Appendix E.3).

We formulate the complexity of WDT as the sum of edge lengths in its dual power diagram. Formally, we can write the regularization as follows,

$$L_{weight} = \sum_{i=1,...,N} Length(E_i),$$

where $E_i$ are the edges in the dual power diagram, and $N$ is the number of edges.

### C.5 Real Regularization

Real regularization is a regularization that is used for maintaining the real values of the connected points in WDT as similar as possible. Also, we leverage this regularization to make real values of points that are connected to the points with high real values to become higher, so that they can be considered in reconstruction more often than the points that are not connected to those points. To be specific, note that we ignore faces with very low existence probability in the reconstruction process. By using this regularization, it can remove holes more effectively.

This real regularzation can be described as

$$L_{real} = \frac{1}{\sum_{i=1,...,N} \Lambda(F_i)} \sum_{i=1,...,N} \Lambda(F_i) \cdot (\sigma_1(F_i) + \sigma_2(F_i)),$$

$$\sigma_1(F_i) = \frac{1}{3} \sum_{j=1,2,3} |\psi_j - \frac{(\psi_1 + \psi_2 + \psi_3)}{3}|,$$

$$\sigma_1(F_i) = \frac{1}{3} \sum_{j=1,2,3} |1 - \psi_j| \cdot \mathbb{I}(\max_{j=1,2,3}(\psi_j) > \delta_{high}).$$

Here $\psi_{1,2,3}$ represent the real values of points that comprise $F_i$, and $\delta_{high}$ is a threshold to determine "high" real value, which is set as 0.8 in our experiments. Note that the faces with higher existence probabilities are prioritized over the others.

## C.6 Quality Regularization

After reconstruction, we usually want to have a mesh that is comprised of triangles of good quality, rather than ill-formed triangles. We adopt the aspect ratio as a quality measure for the triangular faces, and minimize the sum of aspect ratios for all faces during optimization to get a mesh of good quality. Therefore, we can write the regularization as follows.

$$L_{qual} = \frac{1}{\sum_{i=1,...,N} \Lambda(F_i)} \sum_{i=1,...,N} AR(F_i) \cdot E_{max}(F_i) \cdot \Lambda(F_i),$$

$$AR(F_i) = \frac{E_{max}(F_i)}{H_{min}(F_i)} \cdot \frac{\sqrt{3}}{2},$$

$$E_{max}(F_i) = \text{Maximum edge length of } F_i,$$

$$H_{min}(F_i) = \text{Minimum height of } F_i.$$

Note that we prioritize faces with larger maximum edge length and higher existence probability than the others in this formulation. In Appendix E.3, we provide ablation studies for this regularization.

# D Optimization Process

In this section, we explain the optimization processes, or exact reconstruction algorithms, in detail. First, we discuss the optimization process for the experiment in Section 5.1, where we represent the ground truth mesh with DMesh. Then, we discuss the overall optimization process for point cloud or multi-view reconstruction tasks in Section 5.2, from initialization to post processing.

## D.1 Mesh to DMesh

Our overall algorithm to convert the ground truth mesh into DMesh is outlined in Algorithm 1. We explain each step in detail below.

### D.1.1 Point Initialization

At the start of optimization, we initialize the point positions ($\mathbb{P}$), weights ($\mathbb{W}$), and real values ($\psi$) using the given ground truth information ($\mathbb{P}_{gt}$, $\mathbb{F}_{gt}$). To be specific, we initialize the point attributes as follows.

$$\mathbb{P} = \mathbb{P}_{gt}, \quad \mathbb{W} = [1, ..., 1], \quad \psi = [1, ..., 1].$$

The length of vector $\mathbb{W}$ and $\psi$ is equal to the number of points. In Figure 14, we illustrate the initialized DMesh using these point attributes, which becomes the convex hull of the ground truth mesh.

**Algorithm 1** Mesh to DMesh

$\mathbb{P}_{gt}, \mathbb{F}_{gt} \leftarrow$ Ground truth mesh vertices and faces
$\mathbb{P}, \mathbb{W}, \psi \leftarrow$ Initialize point attributes for DMesh
$\bar{\mathbb{F}} \leftarrow$ Empty set of faces
**while** *Optimization not ended* **do**
    $\mathbb{P}, \mathbb{W}, \psi \leftarrow$ Do point insertion, with $\mathbb{P}, \bar{\mathbb{F}}$
    $WDT, PD \leftarrow$ Run WDT algorithm, with $\mathbb{P}, \mathbb{W}$
    $\bar{\mathbb{F}} \leftarrow$ Update faces to exclude, with $WDT$
    $\Lambda(\mathbb{F}_{gt}), \Lambda(\bar{\mathbb{F}}) \leftarrow$ Compute existence probability for faces, with $\mathbb{P}, \psi, WDT, PD$
    $L_{recon} \leftarrow$ Compute reconstruction loss, with $\Lambda(\mathbb{F}_{gt}), \Lambda(\bar{\mathbb{F}})$
    Update $\mathbb{P}, \mathbb{W}, \psi$ to minimize $L_{recon}$
    Bound $\mathbb{P}$
**end**
$M \leftarrow$ Get final mesh from DMesh

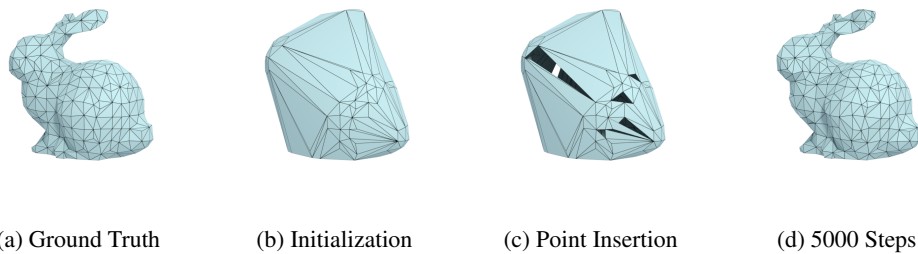

(a) Ground Truth      (b) Initialization      (c) Point Insertion      (d) 5000 Steps

Figure 14: **Intermediate results in converting bunny model to DMesh.** For given ground truth mesh in (a), we initialize our point attributes using the mesh vertices. (b) Then, the initial mesh becomes convex hull of the original mesh. (c) To remove undesirable faces that were not in the original mesh, we insert additional points on the undesirable faces. Then, some of them disappear because of the inserted points. (d) After optimizing 5000 steps, just before another point insertion, DMesh recovers most of the ground truth connectivity.

Note that during optimization, we allow only small perturbations to the positions of initial points, and fix weights and real values of them to 1. This is because we already know that these points correspond to the ground truth mesh vertices, and thus should be included in the final mesh without much positional difference. In our experiments, we set the perturbation bound as 1% of the model size.

However, we notice that we cannot restore the mesh connectivity with only small perturbations to the initial point positions, if there are no additional points that can aid the process. Therefore, we periodically perform point insertion to add additional points, which is described below.

### D.1.2   Point Insertion

The point insertion is a subroutine to add additional points to the current point configurations. It is performed periodically, at every fixed step. The additional points are placed at the random place on the faces in $\bar{\mathbb{F}}$, which correspond to the faces that should not exist in the final mesh. Therefore, these additional points can aid removing these undesirable faces.

However, we found out that inserting a point for every face in $\bar{\mathbb{F}}$ can be quite expensive. Therefore, we use $k$-means clustering algorithm to aggregate them into $0.1 \cdot N_F$ clusters, where $N_F$ is the number of faces in $\bar{\mathbb{F}}$, to add the centroids of the clusters to our running point set. On top of that, we select 1000 random faces in $\bar{\mathbb{F}}$ to put additional points directly on them. This is because there are cases where centroids are not placed on the good positions where they can remove the undesirable faces.

In Figure 14, we render DMesh after point insertion to the initialized mesh. Note that some of the undesirable faces disappear because of the added points.

**Algorithm 2** Point cloud & Multi-view Reconstruction

---

$T \leftarrow$ Observation (Point cloud, Multi-view images)
$\mathbb{P}, \mathbb{W}, \psi \leftarrow$ Initialize point attributes for DMesh (using T if possible)
$\mathbb{F} \leftarrow$ Empty set of faces
**while** *epoch not ended* **do**
    $\mathbb{P}, \mathbb{W}, \psi \leftarrow$ (If not first epoch) Initialize point attributes with sample points from current DMesh, for mesh refinement
    // Phase 1
    **while** *step not ended* **do**
        $WDT, PD \leftarrow$ Run WDT algorithm with $\mathbb{P}, \mathbb{W}$
        $\mathbb{F} \leftarrow$ Update faces to evaluate existence probability for, with $WDT$
        $\Lambda(\mathbb{F}) \leftarrow$ Compute existence probability for faces in $\mathbb{F}$, with $\mathbb{P}, \psi, WDT, PD$
        $L_{recon} \leftarrow$ Compute reconstruction loss, with $\mathbb{P}, \mathbb{F}, \Lambda(\mathbb{F}), T$
        $L_{weight} \leftarrow$ Compute weight regularization, with $PD$
        $L_{real} \leftarrow$ Compute real regularization, with $\mathbb{P}, \psi, WDT$
        $L_{qual} \leftarrow$ Compute quality regularization, with $\mathbb{P}, \mathbb{F}, \Lambda(\mathbb{F})$
        $L \leftarrow L_{recon} + \lambda_{weight} \cdot L_{weight} + \lambda_{real} \cdot L_{real} + \lambda_{qual} \cdot L_{qual}$
        Update $\mathbb{P}, \mathbb{W}, \psi$ to minimize $L$
    **end**
    // Phase 2
    $WDT, PD \leftarrow$ Run WDT algorithm with $\mathbb{P}, \mathbb{W}$
    $\mathbb{F} \leftarrow$ Faces in $WDT$
    $\Lambda_{wdt}(\mathbb{F}) \leftarrow 1$
    **while** *step not ended* **do**
        $\Lambda(\mathbb{F}) \leftarrow$ Compute existence probability for $\mathbb{F}$, with $\mathbb{P}, \psi, \Lambda_{wdt}(\mathbb{F})$
        $L_{recon} \leftarrow$ Compute reconstruction loss, with $\mathbb{P}, \mathbb{F}, \Lambda(\mathbb{F}), T$
        $L_{real} \leftarrow$ Compute real regularization, with $\mathbb{P}, \psi, WDT$
        $L \leftarrow L_{recon} + \lambda_{real} \cdot L_{real}$
        Update $\psi$ to minimize $L$
    **end**
**end**
$M \leftarrow$ Get final mesh from DMesh, after post-processing

---

### D.1.3   Maintaining $\bar{\mathbb{F}}$

In this problem, we minimize the reconstruction loss specified in Eq. 15 to restore the connectivity in the ground truth mesh, and remove faces that do not exist in it. In the formulation, we denoted the faces that are comprised of mesh vertices $\mathbb{P}$, but are not included in the original mesh as $\bar{\mathbb{F}}$. Even though we can enumerate all of them, the total number of faces in $\bar{\mathbb{F}}$ mounts to $O(N^3)$, where $N$ is the number of mesh vertices. Therefore, rather than evaluating all of those cases, we maintain a set of faces $\bar{\mathbb{F}}$ that we should exclude in our mesh during optimization.

To be specific, at each iteration, we find faces in the current WDT that are comprised of points in $\mathbb{P}$, but do not exist in $\mathbb{F}$, and add them to the running set of faces $\bar{\mathbb{F}}$. On top of that, at every pre-defined number of iterations, in our case 10 steps, we compute $k$-nearest neighboring points for each point in $\mathbb{P}$. Then, we find faces that can be generated by combining each point with 2 of its $k$-nearest points, following Rakotosaona u. a. (2021). Then, we add the face combinations that do not belong to $\mathbb{F}$ to $\bar{\mathbb{F}}$. In our experiments, we set $k = 8$.

### D.2   Point cloud & Multi-view Reconstruction

In Algorithm 2, we describe the overall algorithm that is used for point cloud and multi-view reconstruction tasks. We explain each step in detail below.

### D.2.1   Two Phase Optimization

We divide each optimization epoch in two phases. In the first phase (phase 1), we optimize all of the point attributes – positions, weights, and real values. However, in the second phase (phase 2), we fix the point positions and weights, and only optimize the real values.

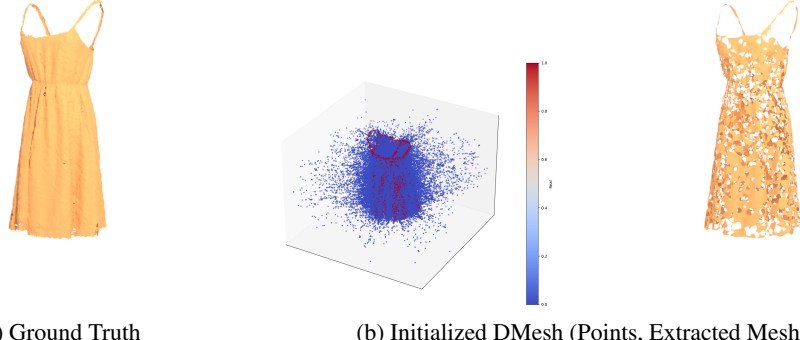

(a) Ground Truth                    (b) Initialized DMesh (Points, Extracted Mesh)

Figure 15: **Initialized DMesh using sample points from ground truth mesh.** (a) From ground truth mesh, we uniformly sample $10K$ points to initialize DMesh. (b) In the left figure, sample points from the ground truth mesh ($\mathbb{P}_{sample}$) are rendered in red. The points that correspond to $\mathbb{P}_{voronoi}$ are rendered in blue. In the right figure, we render the initial mesh we can get from the points, which has a lot of holes.

This design aims at removing ambiguity in our differentiable formulation. That is, even though we desire face existence probabilities to converge to either 0 and 1, those probabilities can converge to the values in between. To alleviate this ambiguity, after the first phase ends, we fix the tessellation to make $\Lambda_{wdt}$ for each face in $\mathbb{F}$ to either 0 or 1. Therefore, in the second phase, we only care about the faces that exist in current $WDT$, which have $\Lambda_{wdt}$ value of 1. Then, we can only care about real values.

Note that the two differentiable renderers that we introduced in Appendix C.3 are designed to serve for these two phases, respectively.

### D.2.2 Point Initialization with Sample Points

In this work, we propose two point initialization methods. The first initialization method can be used when we have sample points near the target geometry in hand.

This initialization method is based on an observation that the vertices of Voronoi diagram of a point set tend to lie on the medial axis of the target geometry (Amenta u. a., 1998a,b). Therefore, for the given sample point set $\mathbb{P}_{sample}$, we first build Voronoi diagram of it, and find Voronoi vertices $\mathbb{P}_{voronoi}$. Then, we merge them to initialize our point set $\mathbb{P}$:

$$\mathbb{P} = \mathbb{P}_{sample} \cup \mathbb{P}_{voronoi},$$

all of which weights are initialized to 1. Then, we set the real values ($\psi$) of points in $\mathbb{P}_{sample}$ as 1, while setting those of points in $\mathbb{P}_{voronoi}$ as 0.

In Figure 15, we render the mesh that we can get from this initialization method, when we use $10K$ sample points. Note that the initial mesh has a lot of holes, because there could be Voronoi vertices that are located near the mesh surface, as pointed out by (Amenta u. a., 1998b). However, we can converge to the target mesh faster than the initialization method that we discuss below, because most of the points that we need are already located near the target geometry.

### D.2.3 Point Initialization without Sample Points

If there is no sample point that we can use to initialize our points, we initialize our points with $N^3$ points regularly distributed on a grid structure that encompasses the domain, all of which has weight 1 and $\psi$ value of 1. We set $N = 20$ for every experiment (Figure 16a). Then, we optimize the mesh to retrieve a coarse form of the target geometry (Figure 16b). Note that we need to refine this mesh in the subsequent epochs, as explained below.

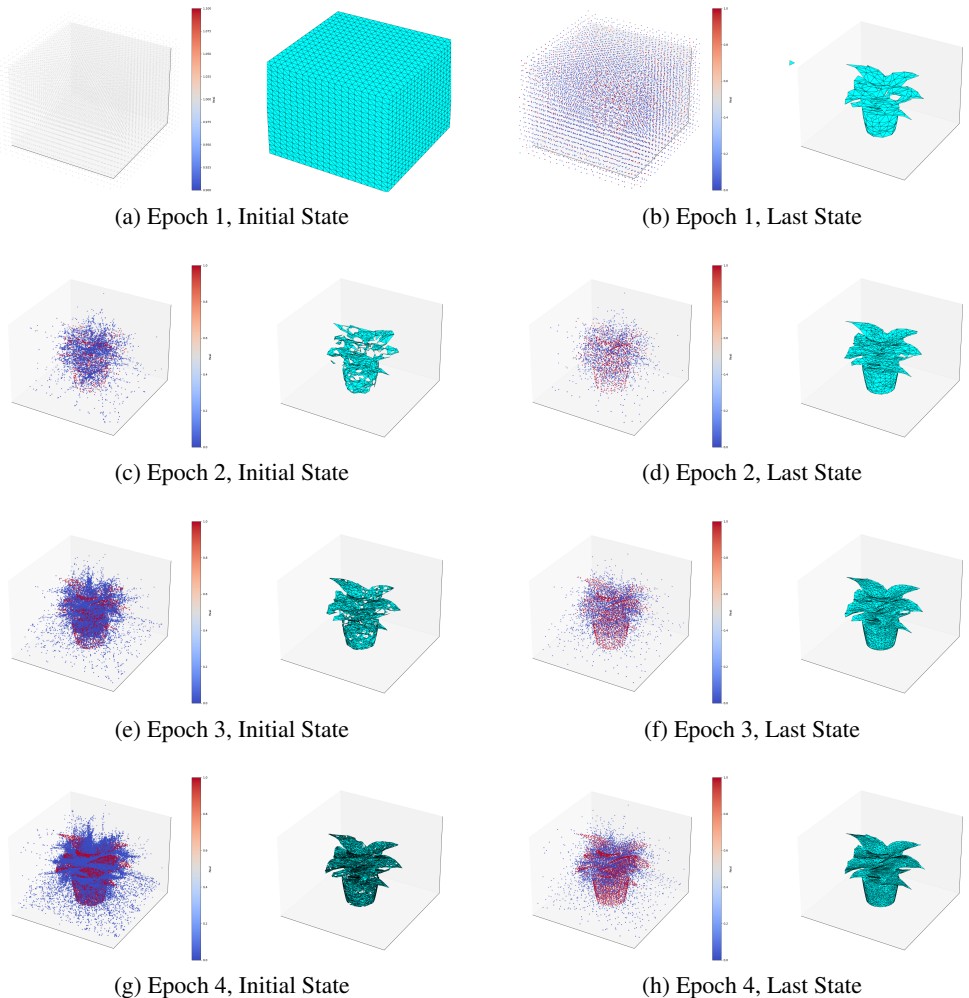

(a) Epoch 1, Initial State

(b) Epoch 1, Last State

(c) Epoch 2, Initial State

(d) Epoch 2, Last State

(e) Epoch 3, Initial State

(f) Epoch 3, Last State

(g) Epoch 4, Initial State

(h) Epoch 4, Last State

Figure 16: **Optimization process for multi-view reconstruction for Plant model.** At each row, we present the initial state (left) and the last state (right) of each epoch. For each figure, the left rendering shows the point attributes color coded based on real values, while the right one shows the extracted mesh. (a), (b) In the first epoch, we initialize DMesh without sample points. At the end of each epoch, we sample points from the current mesh, and use them for initialization in the next epoch.

### D.2.4    Point Initialization for Different Inputs

Until now, we introduced two point initialization techniques. When the input is a point cloud, we sample subset of the point cloud to initialize our mesh (Figure 15). However, when the input is multi-view images, we start from initialization without sample points (Figure 16), because there is no sample point cloud that we can make use of.

### D.2.5    Maintaining $\mathbb{F}$

We maintain the running set of faces to evaluate probability existence for in $\mathbb{F}$. At each iteration, after we get $WDT$, we insert every face in $WDT$ to $\mathbb{F}$, as it has a high possibility to persist in the subsequent optimization steps. Also, as we did int mesh to DMesh conversion (Appendix D.1), at every 10 optimization step, we find $k$-nearest neighbors for each point, and form face combinations based on them. Then, we add them to $\mathbb{F}$.

### D.2.6 Mesh Refinement

At start of each epoch, if it is not the first epoch, we refine our mesh by increasing the number of points. To elaborate, we refine our mesh by sampling $N$ number of points on the current DMesh, and then initialize point attributes using those sample points as we explained above. We increase $N$ as number of epoch increases. For instance, in our multi-view reconstruction experiments, we set the number of epochs as 4, and set $N = (1K, 3K, 10K)$ for the epochs excluding the first one. In Figure 16, we render the initial and the last state of DMesh of each epoch. Note that the mesh complexity increases and becomes more accurate as epoch proceeds, because we use more points. Therefore, this approach can be regarded as a coarse-to-fine approach.

### D.2.7 Post-Processing

When it comes to multi-view reconstruction, we found out that it is helpful to add one more constraint in defining the face existence. In our formulation, in general, a face $F$ has two tetrahedra $(T_1, T_2)$ that are adjacent to each other over the face. Then, we call the remaining point of $T_1$ and $T_2$ that is not included in $F$ as $P_1$ and $P_2$. Our new constraint requires at least one of $P_1$ and $P_2$ to have $\psi$ value of 0 to let $F$ exist.

This additional constraint was inspired by the fact that $F$ is not visible from outside if $F$ exists in our original formulation, and both of $P_1$ and $P_2$ have $\psi$ value of 1. That is, if it is not visible from outside, we do not recognize its existence. This constraint was also adopted to accommodate our real regularization, which increases the real value of points near surface. If this regularization makes the real value of points inside the closed surface, they would end up in internal faces that are invisible from outside. Because of this invisibility, our loss function cannot generate a signal to remove them. In the end, we can expect all of the faces inside a closed surface will exist, because of the absence of signal to remove them. Therefore, we choose to remove those internal faces by applying this new constraint in the post-processing step.

Note that this discussion is based on the assumption that our renderer does a precise depth testing. If it does not do the accurate depth testing, internal faces can be regarded as visible from outside, and thus get false gradient signal. In Figure 13a, the final mesh after phase 1 is rendered, and we can see therer are lots of internal faces as the renderer used in phase 1 does not support precise depth testing. However, we can remove them with the other renderer in phase 2, as shown in Figure 13b, which justifies our implementation of two different renderers.

Finally, we note that this constraint is not necessary for point cloud reconstruction, because if we minimize $CD_{ours}$ in Appendix C.2, the internal faces will be removed automatically.

## E Experimental Details

In this section, we provide experimental details for the results in Section 5, and visual renderings of the our reconstructed mesh. Additionally, we provide the results of ablation studies about regularizations that we suggested in Section 4.3.

### E.1 Mesh to DMesh

As shown in Table 1, we reconstruct the ground truth connectivity of Bunny, Dragon, and Buddha model from Stanford dataset (Curless und Levoy, 1996). For all these experiments, we optimized for $20K$ steps, and used an ADAM optimizer (Kingma und Ba, 2014) with learning rate of $10^{-4}$. For Bunny model, we inserted additional points at every 5000 step. For the other models, we inserted them at every 2000 step.

In Figure 17, we provide the ground truth mesh and our reconstructed mesh. We can observe that most of the connectivity is preserved in our reconstruction, as suggested numerically in Table 1. However, note that the appearance of the reconstructed mesh can be slightly different from the ground truth mesh, because we allow $1\%$ of positional perturbations to the mesh vertices.

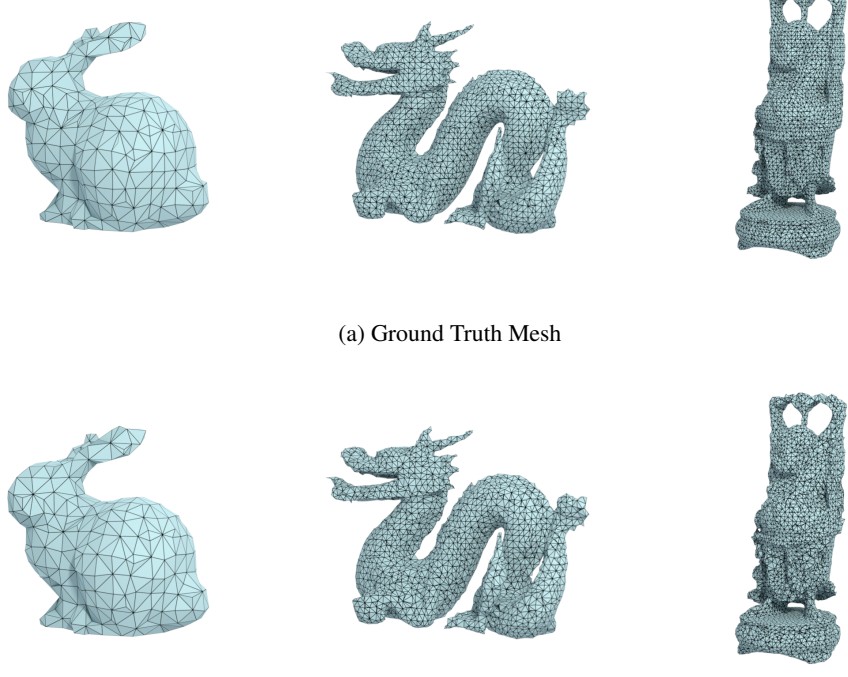

(a) Ground Truth Mesh

(b) Reconstructed DMesh

Figure 17: **Reconstruction results for mesh to DMesh experiment. From Left: Bunny, Dragon, and Buddha.** We can observe that most of the edge connectivity is perserved in the reconstruction, even though the appearance is slightly different from the ground truth mesh because of small perturbations of vertex positions.

## E.2 Point Cloud & Multi-view Reconstruction

### E.2.1 Hyperparameters for Point Cloud Reconstruction

- Optimizer: ADAM Optimizer, Learning rate = $10^{-4}$ for open surface meshes and two mixed surface meshes (Bigvegas, Raspberry) / $3 \cdot 10^{-4}$ for closed surface meshes, and one mixed surface mesh (Plant).
- Regularization: $\lambda_{weight} = 10^{-8}, \lambda_{real} = 10^{-3}, \lambda_{qual} = 10^{-3}$ for every mesh.
- Number of epochs: Single epoch for every mesh.
- Number of steps per epoch: 1000 steps for phase 1, 500 steps for phase 2 for every mesh.

### E.2.2 Hyperparameters for Multi-view Reconstruction

- Optimizer: ADAM Optimizer, Learning rate = $10^{-3}$ in the first epoch, and $3 \cdot 10^{-4}$ in the other epochs for every mesh.
- Weight Regularization: $\lambda_{weight} = 10^{-8}$ for every mesh.
- Real Regularization: $\lambda_{real} = 10^{-3}$ for the first 100 steps in every epoch for open surface meshes and one mixed surface mesh (Plant) / $10^{-2}$ for the first 100 steps in every epoch for closed surface meshes and two mixed surface meshes (Bigvegas, Raspberry).
- Quality Regularization: $\lambda_{qual} = 10^{-3}$ for every mesh.
- Normal Coefficient: $\lambda_{normal} = 0$ for every mesh (Eq. 17).
- Number of epochs: 4 epochs for every mesh. In the first epoch, use $20^{-3}$ regularly distributed points for initialization. In the subsequent epochs, sample $1K, 3K$, and $10K$ points from the current mesh for initialization.

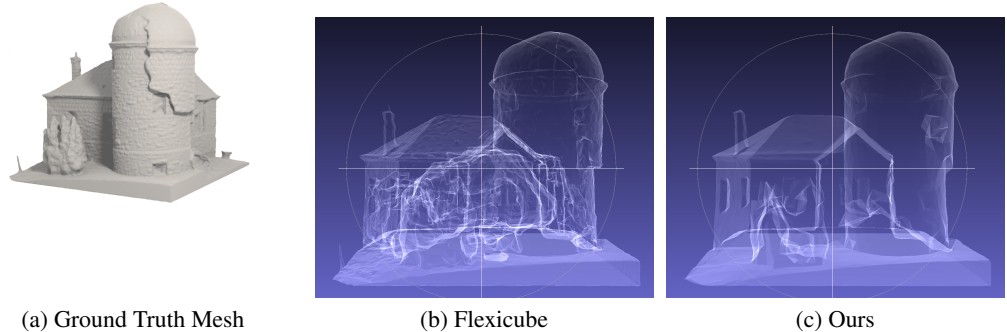

| (a) Ground Truth Mesh | (b) Flexicube | (c) Ours |

Figure 18: **Reconstruction results for a closed surface model in Thingi32 dataset.** Flexi-cube (Shen u. a., 2023) can generate internal structures, while our approach removes them through post-processing.

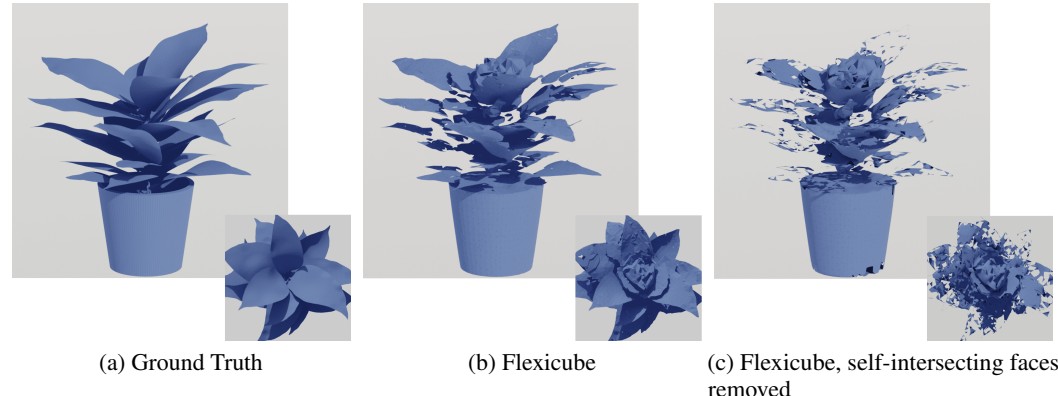

| (a) Ground Truth | (b) Flexicube | (c) Flexicube, self-intersecting faces removed |

Figure 19: **Reconstruction results for the Plant model.** Flexicube (Shen u. a., 2023) can generate redundant, self-intersecting faces for open surfaces, in this case, leaves. To better capture the redundant faces, we rendered the models from upper side, which is shown in the bottom right figures.

- Number of steps per epoch: 500 steps for phase 1, 500 steps for phase 2 for every mesh.
- Batch size: 64 for open surface meshes, 16 for the other meshes.

### E.2.3   Visual Renderings

In Figure 22, 23, and 24, we provide visual renderings of our point cloud and multi-view reconstruction results with ground truth mesh. We also provide illustration of input point cloud and diffuse map. Note that we also used depth renderings for multi-view reconstruction experiments.

### E.2.4   Additional Discussion

Generally, we can observe that reconstruction results from both point cloud and multi-view images capture the overall topology well. However, we noticed that the multi-view reconstruction results are not as good as point cloud reconstruction results. In particular, we can observe small holes in the multi-view reconstruction results. We assume that these artifacts are coming from relatively weaker supervision of multi-view images than dense point clouds. Also, we believe that we can improve these multi-view reconstruction results with more advanced differentiable renderer, and better mesh refinement strategy. In the current implementation, we lose connectivity information at the start of each epoch, which is undesirable. We believe that we can improve this approach by inserting points near the regions of interest, rather than resampling over entire mesh.

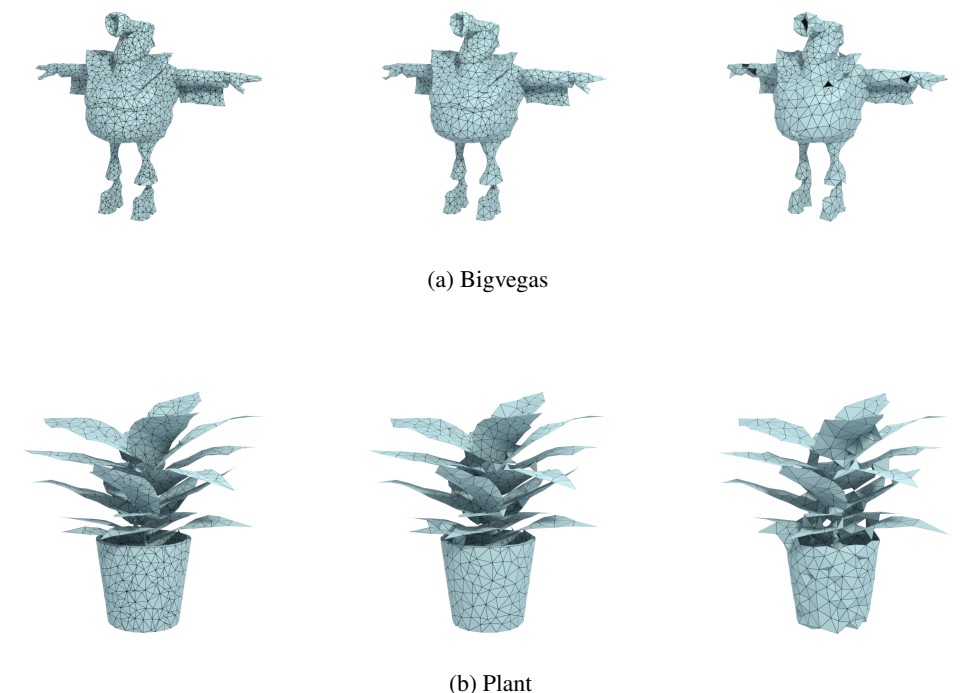

(a) Bigvegas

(b) Plant

Figure 20: **Point cloud reconstruction results with different $\lambda_{weight}$.** From Left: $\lambda_{weight} = 10^{-6}, 10^{-5}$, and $10^{-4}$.

Also, regarding comparison to Flexicube (Shen u. a., 2023) in Table 2, we tried to found out the reason why ours give better results than Flexicube in terms of CD to the ground truth mesh for closed surfaces in thingi32 dataset. We could observe that Flexicube's reconstruction results capture fine geometric details on the surface mesh, but also observed that they have lots of false internal structure (Figure 18). Note that this observation not only applies to closed surfaces, but also to open surfaces, where it generates lots of false, self-intersecting faces (Figure 19). Our results do not suffer from these problems, as we do post-processing (Appendix D.2) to remove inner structure, and also our method can represent open surfaces better than the volumetric approaches without self-intersecting faces.

### E.3 Ablation studies

In this section, we provide ablation studies for the regularizations that we proposed in Section 4.3. We tested the effect of the regularizations on the point cloud reconstruction task.

#### E.3.1 Weight Regularization

We tested the influence of weight regularzation in the final mesh, by choosing $\lambda_{weight}$ in $(10^{-6}, 10^{-5}, 10^{-4})$. Note that we set the other experimental settings as same as described in Section E.2, except $\lambda_{quality}$, which is set as 0, to exclude it from optimization.

In Table 5, we provide the quantitative results for the experiments. For different $\lambda_{weight}$, we reconstructed mesh from point clouds, and computed average Chamfer Distance (CD) and average number of faces across every test data. We can observe that there exists a clear tradeoff between CD and mesh complexity. To be specific, when $\lambda_{weight} = 10^{-6}$, the CD is not very different from the results in Table 2, where we use $\lambda_{weight} = 10^{-8}$. However, when it increases to $10^{-5}$ and $10^{-4}$, we can observe that the mesh complexity (in terms of number of faces) decreases, but CD increases quickly.

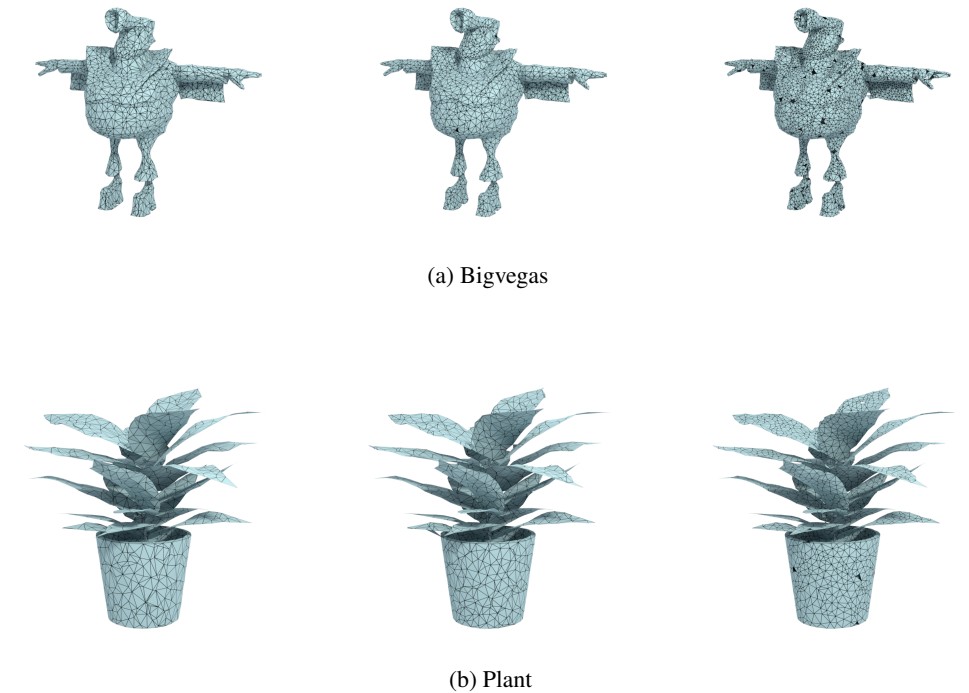

(a) Bigvegas

(b) Plant

Figure 21: **Point cloud reconstruction results with different** $\lambda_{quality}$. From Left: $\lambda_{real} = 10^{-4}, 10^{-3}$, and $10^{-2}$.

The renderings in Figure 20 support these quantitative results. When $\lambda_{weight} = 10^{-6}$, we can observe good reconstruction quality. When $\lambda_{weight} = 10^{-5}$, there are small artifacts in the reconstruction, but we can get meshes of generally good quality with fewer number of faces. However, when it becomes $10^{-4}$, the reconstruction results

Table 5: Ablation study for weight regularization, quantitative results.

| $\lambda_{weight}$ | $10^{-6}$ | $10^{-5}$ | $10^{-4}$ |
|---|---|---|---|
| CD | 7.48 | 8.08 | 10.82 |
| Num. Face | 4753 | 2809 | 1786 |

deteriorate, making holes and bumpy faces on the smooth surface. Therefore, we can conclude that weight regularization contributes to reducing the mesh complexity. However, we need to choose $\lambda_{weight}$ carefully, so that it does not harm the reconstruction quality. The experimental results tell us setting $\lambda_{weight}$ to $10^{-6}$ could be a good choice to balance between these two contradictory objectives.

### E.3.2 Quality Regularization

As we did in the previous section, we test the influence of quality regularization in the final mesh by selecting $\lambda_{real}$ among $(10^{-4}, 10^{-3}, 10^{-2})$. We also set the other experimental settings as same as before, except $\lambda_{weight} = 0$.

In Table 6 and Figure 21, we present quantitative and qualitative comparisons between the reconstruction results. We provide statistics about average CD, average number of faces, and average aspect ratio of faces. Interestingly, unlike weight regularization, we could not observe tradeoff between CD and aspect ratio. Rather than that, we could find that CD decreases as aspect ratio gets smaller, and thus the triangle quality gets better.

Table 6: Ablation study for quality regularization, quantitative results.

| $\lambda_{qual}$ | $10^{-4}$ | $10^{-3}$ | $10^{-2}$ |
|---|---|---|---|
| CD | 7.60 | 7.42 | 7.28 |
| Num. Face | 8266 | 8349 | 10806 |
| Aspect Ratio | 2.33 | 2.06 | 1.55 |

We find the reason for this phenomenon in the increase of smaller, good quality triangle faces. Note that there is no significant difference between the number of faces between $\lambda_{qual} = 10^{-4}$ and $10^{-3}$. Also, we cannot find big difference between visual renderings between them, even though the aspect

ratio was clearly improved. However, when $\lambda_{qual}$ becomes $10^{-2}$, the number of faces increase fast, which can be observed in the renderings, too. We believe that this increase stems from our quality constraint, because it has to generate more triangles to represent the same area, if there is less degree of freedom to change the triangle shape. Since it has more triangle faces, we assume that they contribute to capturing fine details better, leading to the improved CD.

However, at the same time, note that the number of holes increase as we increase $\lambda_{qual}$, which lead to visual artifacts. We assume that there are not enough points to remove these holes, by generating quality triangle faces that meet our needs. Therefore, as discussed before, if we can find a systematic way to prevent holes, or come up with a better optimization scheme to remove them, we expect that we would be able to get accurate mesh comprised of better quality triangles.

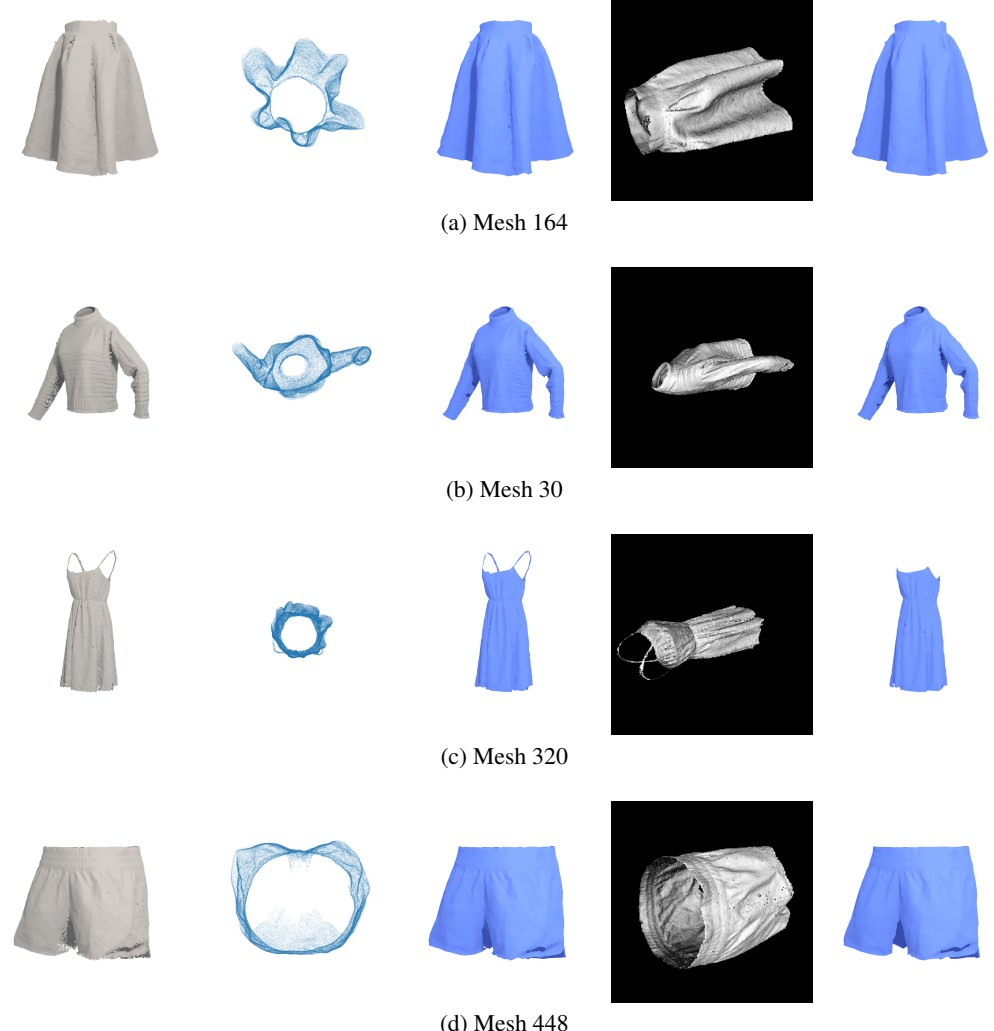

(a) Mesh 164

(b) Mesh 30

(c) Mesh 320

(d) Mesh 448

Figure 22: **Point cloud and Multi-view Reconstruction results for open surface models.** From Left: Ground truth mesh, sample point cloud, point cloud reconstruction results, diffuse rendering, multi-view reconstruction results.

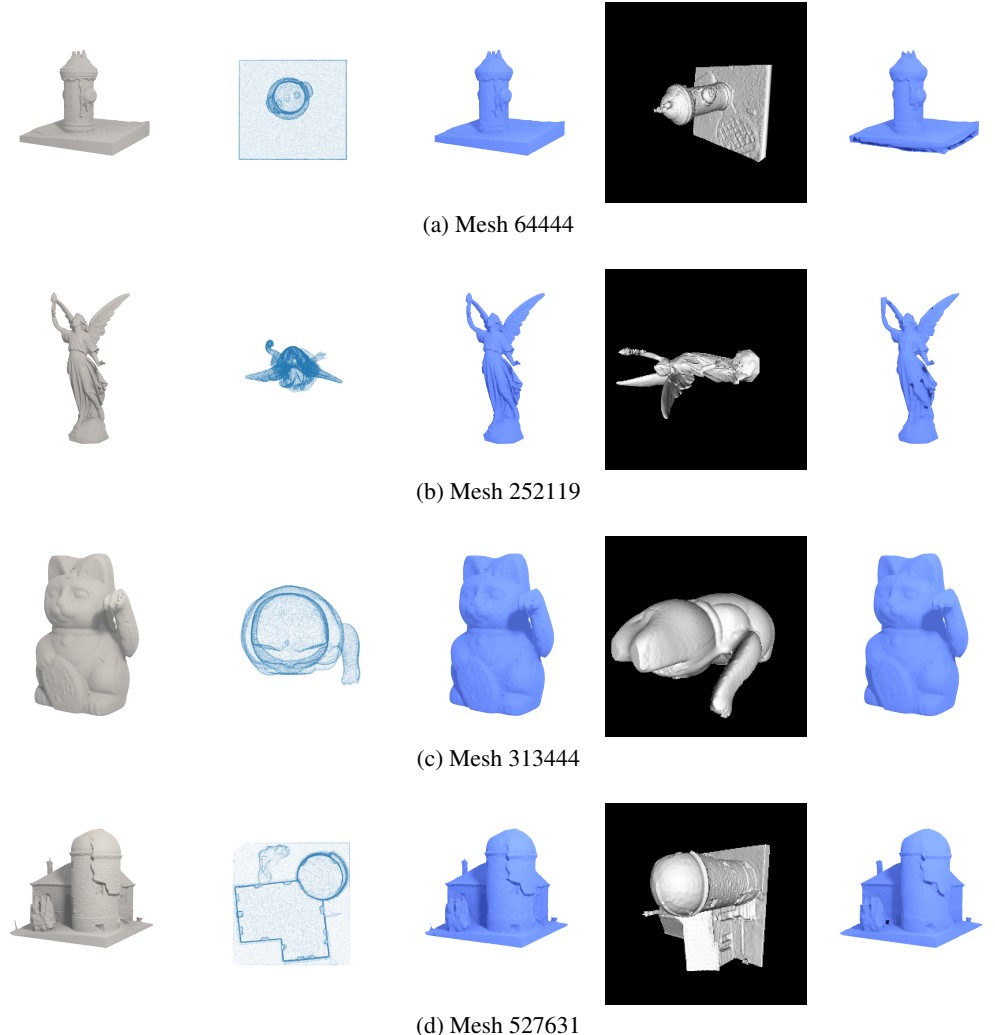

(a) Mesh 64444

(b) Mesh 252119

(c) Mesh 313444

(d) Mesh 527631

Figure 23: **Point cloud and Multi-view Reconstruction results for closed surface models.** From Left: Ground truth mesh, sample point cloud, point cloud reconstruction results, diffuse rendering, multi-view reconstruction results.

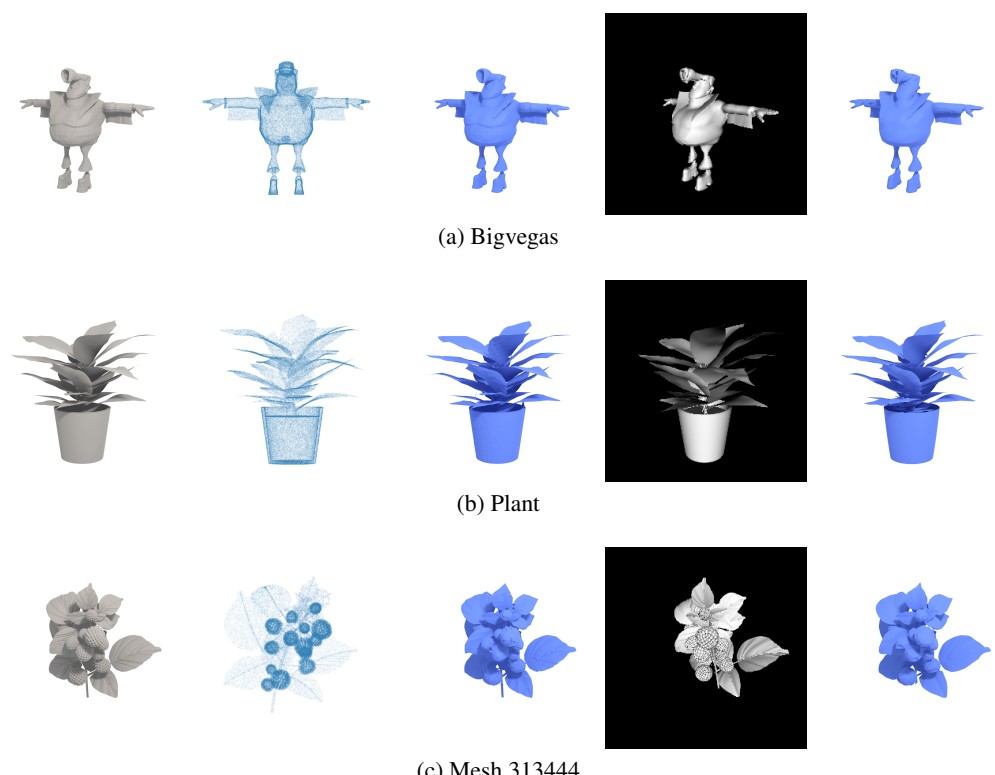

(a) Bigvegas

(b) Plant

(c) Mesh 313444

Figure 24: **Point cloud and Multi-view Reconstruction results for mixed surface models.** From Left: Ground truth mesh, sample point cloud, point cloud reconstruction results, diffuse rendering, multi-view reconstruction results.

