# OpenReview forum: "DMesh: A Differentiable Mesh Representation"
_NeurIPS.cc/2024/Conference — NeurIPS 2024 poster_

### Official Review · Reviewer_7cDg · 2024-07-12

**Soundness:** 4
**Presentation:** 3
**Contribution:** 4
**Rating:** 7
**Confidence:** 4

**Summary:**

This paper proposed a novel differentiable representation of a mesh. It takes advantage of two probabilities, $\Lambda_{wdt}$ and $\Lambda_{real}$, to generate the mesh, such that the mesh is differentiable.

**Strengths:**

Based on weighted Delaunay triangulation and its dual power diagram, the paper proposed a novel approach to compute $\Lambda_{wdt}$, which makes the WDT generation differentiable.

It seems that the accuracy of the conversion from mesh to DMesh is high enough. The paper applied the novel representation for various applications, such as, point cloud and multi-view reconstruction.

**Weaknesses:**

The efficiency of the method seems some what slow due to the WDT construction.

In the experiments, it is better to show the deformation process of the a mesh with different topologies.

**Questions:**

As there are spurious non-manifold errors in the mesh, what are the ratio of the non-manifold errors?

Is the method robust against noises and holes? Current experiments did not test point clouds with noises or holes.

**Limitations:**

The limitations should discuss the potential limits to process indoor/outdoor scenes. These data are more challenge due to noises, holes and complicated topology.

---

> ### Author Rebuttal · Authors · 2024-08-06
>
> We appreciate your detailed comments and positive evaluation about our work. Please let us know if we addressed your questions correctly.
>
> -----------------------------------------------
> **Q1. Degraded efficiency of the method due to WDT**
>
> A1. As pointed out, the main computational bottleneck in the current implementation is the WDT construction. Please see Global Response A1 for detailed information. Removing the necessity for WDT is one of our future research directions.
>
> -----------------------------------------------
> **Q2. Showing the deformation process of a mesh with different topologies**
>
> A2. Thank you for the suggestion. Please see Global Response A3. We will include this in the revised version of the paper.
>
> -----------------------------------------------
> **Q3. Ratio of non-manifold errors**
>
> A3. Please see Global Response A2 for details on the ratio of non-manifold errors.
>
> -----------------------------------------------
> **Q4. Robustness against noises and holes in point cloud reconstruction**
>
> A4. As pointed out, we did not test our method on point clouds with noises or holes. Currently, our method may not be suitable for such challenging cases because our reconstruction is driven solely by Chamfer Distance loss. Consequently, holes or noises in the input point clouds may persist in the reconstructed mesh. Other point cloud reconstruction methods often impose assumptions about the mesh topology to address these issues. However, our approach does not impose any topological constraints, which is contradictory to these methods.
>
> Additionally, developing a robust point cloud reconstruction method is beyond the scope of this paper, as the main contribution lies in introducing the differentiable mesh formulation, not the reconstruction itself. Nevertheless, we acknowledge the importance of this aspect and consider it an interesting direction for future research.
>
> -----------------------------------------------
> **Q5. Limitations in processing indoor/outdoor scenes**
>
> A5. Yes, we will include a discussion of the limitations related to processing indoor and outdoor scenes in the paper, as these data are more challenging due to noises, holes, and complicated topology. Thank you for the suggestion.

---

> > ### Comment · Reviewer_7cDg · 2024-08-12
> >
> > Thank you for your responses. I choose to keep my rating.

---

> > > ### Author Response · Authors · 2024-08-12
> > >
> > > Thank you for your acknowledgement again, we really appreciate it.
> > >
> > > Authors

---

### Official Review · Reviewer_jRqE · 2024-07-12

**Soundness:** 3
**Presentation:** 3
**Contribution:** 3
**Rating:** 6
**Confidence:** 3

**Summary:**

This paper proposes a novel differentiable mesh representation, which focuses on both the vertex position and connectivity. To achieve this, authors come from the probability of face existence, and decompose the probability into two parts: 1) the WDT probability; 2) a probability of vertices' existence on surface. Further, to accelerate the WDT process, authors adopt a novel relaxation operation, making the whole design applicable.

**Strengths:**

+: The probabilistic approach to determining the existence of faces in the mesh is innovative and allows for gradient-based optimization of both vertex positions and their connectivity, which reveals the potential application in many optimization or learning-based reconstruction tasks.

+: Both the efficient CUDA implementation and the relaxation design further enhance the usability.

+: Sufficient evaluations on settings: mesh reconstruction from ground truth, point clouds and multi-view images, verifies the effectiveness of the paper. And the quantitative comparisons with other methods are well-presented and solid.

**Weaknesses:**

-: Although the method is efficient compared to traditional approaches, the computational cost is still significant, especially when dealing with large datasets. This prevents the application beyond small object level task.

-: While the method is tested on synthetic datasets and controlled environments, there is a lack of demonstration on real-world datasets, or texture-rich dataset. Applying DMesh to more complex scenarios would strengthen the paper's contributions.

-: No ablation on the influence of \lambda_WDT and \lambda_real.

**Questions:**

I think this is a good work and I believe further improvements as mentioned in the weakness are not easy to complete during the rebuttal. I suggest the author to add experiment on ablation of the influence of \lambda_WDT and \lambda_real to make the core contribution more convincing.

**Limitations:**

Yes, authors discussed limitations.

---

> ### Author Rebuttal · Authors · 2024-08-06
>
> We appreciate your detailed comments and positive evaluation about our work. Please let us know if we addressed your questions correctly.
>
> ------------------------------------
> **Q1. Significant computational cost**
>
> A1. Yes, in the current implementation, we can handle up to 20K points efficiently, but going beyond that is limited. Please see the detailed analysis about the computational cost in Global Response A1. To overcome this limitation, we need to take a completely different approach that does not require the WDT algorithm, as it is the main computational bottleneck. This is one of our future research directions.
>
> ------------------------------------
> **Q2. Testing on real-world and texture-rich datasets**
>
> A2. As pointed out, our current implementation does not support texture reconstruction, which is why our method is not yet applicable to real-world datasets. This is also a future research direction we intend to explore.
>
> ------------------------------------
> **Q3. Ablation on the influence of $\Lambda_{wdt}$ and $\Lambda_{real}$**
>
> A3. We believe there might be a misunderstanding or typo in this question, as $\Lambda_{wdt}$ and $\Lambda_{real}$ are multiplied to get the final face existence probability (Eq. 2). They are not hyperparameters that we can arbitrarily choose. Therefore, there is no specific ablation study we can conduct on them. If you meant $\lambda_{weight}$, $\lambda_{real}$, and $\lambda_{qual}$, which are the coefficients for regularization (Section 4.3.2), please see Figure 7 and Appendix E.3 for the ablation study of these coefficients. Please let us know if we misunderstood your question.

---

> > ### Comment · Reviewer_jRqE · 2024-08-13
> >
> > Thank you for your responses. I choose to keep my rating.

---

> > > ### Author Response · Authors · 2024-08-13
> > >
> > > Thank you again for your acknowledgement!

---

### Official Review · Reviewer_8dk4 · 2024-07-12

**Soundness:** 3
**Presentation:** 3
**Contribution:** 3
**Rating:** 6
**Confidence:** 3

**Summary:**

The paper presents a differentiable mesh representation for mesh reconstruction from input meshes, point clouds, and for multi-view reconstruction. It builds off ideas of Rakotosaona 21, and uses a weighted Delaunay triangulation framework to predict simplex probabilities. They borrow the notion of distance to reduced power cell, but consider lower bound estimations and precomputation to speed it up. Their method is the first to give a differentiable mesh representation that is flexible enough to handle open/closed meshes, and accommodate changes in mesh topology and connectivity. The method is tested against several competing SOTA works and performs better or comparably in all scenarios (save for producing exactly manifold output).

**Strengths:**

1. As noted, the framework is the first to directly represent meshes in a differentiable way that can handle topology change.
2. They extend (in a straightforward way) the framework of Rakotosaona et al, to the setting of surfaces in 3D, and note the possibility of its use for other k,d (though do not demonstrate use in these scenarios).
3. They provide computational speedup methods that help to alleviate the computational burden of power diagram computation.
4. The authors provide extensive validation and implementation details in the supplementary, including some ablation studies.

**Weaknesses:**

1. The method does not guarantee manifoldness of the end mesh result, which is acknowledged by the authors.
2. The matter of computational cost is swept under the rug a bit. It would be nice to present detailed statistics of some sort (over meshes/point clouds of various sizes, etc.) in the main paper.
3. It is unclear to me how much technical novelty is contained in the estimation step of 4.1 over the prior framework. The steps seem relatively natural and easy to consider once you realize that the dual simplex at hand is no longer a simple point.
4. The earlier parts of the article could be compacted to include at least an overview of the technical pipeline used in the experiments. There are a lot of details that are omitted entirely in each use scenario, especially for the multi-view reconstruction.
5. There is a small, confusing technical mistake above equation (3). The signed distance function should be positive inside, and negative when outside for this to make sense. This is as done in Rakotosaona.

**Questions:**

1. As noted I had some technical questions about S4.1:
    * For Eq. (5), can you give an instance when this lower bound is not sharp? I thought that if the simplex does not exist in the WDT, then the power cell would remained unchanged under deletion of the other two points?
    * How is it that one approximates the reduced power cell? This is not explained at all, unless I missed something?
2. Many of the mins,maxes, absolute values, lengths, etc. that appear in the regularization terms appear to be non-differentiable. Are these simply smoothed, or am I missing something?
3. What does the optimization do when there is not a shot showing a particular side of the model? E.g. from the bottom of a model?
4. Why were "reality" weights vertex-based? Was there any consideration for edge-based weights?
5. How often does non-manifoldness occur?

**Limitations:**

The authors appropriately acknowledged their limitations, some of which I listed in Weaknesses above.

---

> ### Author Rebuttal · Authors · 2024-08-06
>
> We appreciate your detailed comments and positive evaluation about our work. Please let us know if we addressed your questions correctly.
>
> ------------------------------
> **Q1. Non-manifoldness**
>
> A1. Please see Global Response A2.
>
> ------------------------------
> **Q2. Computational cost**
>
> A2. Thank you for the suggestion. Please see Global Response A1 for detailed statistics about the computational cost. We will include this information in the final version of the main paper.
>
> ------------------------------
> **Q3. Technical novelty in the estimation step of 4.1**
>
> A3. There were two main motivations for developing the lower bound estimation step in 4.1 instead of using the prior framework: precision and efficiency, as suggested at the end of 3.2. As shown in the paper, our estimations were much more precise than the previous method. Regarding efficiency, our approach significantly reduces the computational burden, *even when the dual simplex is a single point* (Table 4, d=2, k=2 case). *We argue that our approach effectively relaxes the computational challenges of the previous method, despite being based on simple principles*. Additionally, *its implementation was not as straightforward as the prior framework*, which is why we used CUDA instead of PyTorch. For instance, while the prior framework only requires a point projection operation, we needed other operations, such as line-line distance computation. Therefore, *we believe the technical novelty is not trivial*.
>
> ------------------------------
> **Q4. Including an overview of the technical pipeline in the main body**
>
> A4. Thank you for the suggestion. We will modify the paper to incorporate more details about the experiments in the main body if we are granted an additional page in the camera-ready version.
>
> ------------------------------
> **Q5. Technical mistake about description above equation (3)**
>
> A5. Thank you for pointing this out. You are correct that the description was wrong. Specifically, Eq. 3 is correct, but in line 154, we should have written, "sign is positive when inside." We will correct this in the revised version of the paper.
>
> ------------------------------
> **Q6. Sharpness of the lower bound in Eq. (5)**
>
> A6. Thank you for the insightful question. Let's consider an example under the same setting as Figure 6. In Figure 6(d), we rendered the case when $\Delta^{1} = \{ p_1, p_7 \}$. As you pointed out, the reduced power cell of $p_1$ remains unchanged when we delete $p_7$. However, consider the case where $\Delta^{2} = \{ p_1, p_3, p_7 \}$, which also does not exist in the WDT. In this scenario, the reduced power cell of $p_1$ changes when we delete $p_3$ and $p_7$ because $p_1$ and $p_3$ are connected in the power diagram. This observation also applies to our (d=3, k=2) case.
>
> ------------------------------
> **Q7. Approximating the reduced power cell**
>
> A7. We briefly mentioned approximating the reduced power cell with pre-computed PD but did not provide details. To elaborate, as we compute PD at every optimization step, we collect the points that comprise the power cell of a point $P$ at each step. By keeping and accumulating these points, we can effectively find possible half-planes that could comprise the reduced power cell of $P$. Therefore, we use the half-planes between $P$ and the (accumulated list of) points to approximate the (boundary of) reduced power cell of $P$. Although this approximation is not very accurate in the initial optimization steps, it does not undermine the optimization process. This is because, as optimization progresses, most possible half-planes to define the reduced power cell are collected for each point. We will include this detail in the revised version of the paper.
>
> ------------------------------
> **Q8. Non-differentiability of regularization terms**
>
> A8. For min/max functions, we used a smoothed version because they are used to compute face probabilities, a main component of our approach. We found that differentiable min/max functions stabilize the optimization process. However, we used absolute values and lengths without modification because they are almost differentiable and used mainly for regularizers, which did not harm the optimization process.
>
> ------------------------------
> **Q9. Handling unseen parts of the model in multi-view reconstruction**
>
> A9. In multi-view reconstruction tasks, we use real regularization as described in Appendix C.5. This regularization automatically fills unseen parts of the object (e.g., inner parts). In the post-processing step (Appendix D.2.7), we discard most of these automatically generated, invisible faces. Therefore, when a particular side of the model is not depicted, that part would be reconstructed as the maximum volume that does not violate observations from other views. We will include an example of this case in the revised version of the paper.
>
> ------------------------------
> **Q10. Vertex-based reality values**
>
> A10. We did not consider edge or face-based reality values because we aimed to store all mesh information in the points. This design will simplify its adoption in other ML frameworks, such as generative models. It is much easier for neural networks to generate point-wise features than generate unfixed edge or face combinations. By storing all connectivity information in points, we avoid generating face combinations. It is also more efficient since the number of edges or faces becomes excessive as the number of points increases. Storing real values for such edges or faces is overwhelming. However, edge or face-based reality values could be useful in the final optimization step to fine-tune geometry and remove non-manifoldness, as discussed in Global Response A2.

---

> > ### Comment · Reviewer_8dk4 · 2024-08-12
> > **Thank you**
> >
> > Thank you for the extensive responses. I will be keeping my score as is entering the discussion phase.
> >
> > Reviewer

---

> > > ### Author Response · Authors · 2024-08-12
> > >
> > > Thank you for your acknowledgement again, we really appreciate it.
> > >
> > > Authors

---

### Official Review · Reviewer_yemP · 2024-07-12

**Soundness:** 4
**Presentation:** 4
**Contribution:** 4
**Rating:** 8
**Confidence:** 3

**Summary:**

The work introduces a differentiable mesh representation where both the topology of the surface and the connectivity are differentiable. The paper builds on previous work titled 'Differentiable Surface Triangulation,' which suggests using a soft relaxation of Weighted Delaunay Triangulation (WDT) to differentiate through meshing and vertex positions. DMesh highlights that the original work cannot be used as a differentiable mesh representation in a straightforward way and would be very inefficient due to the quadratic scaling of the time complexity with the number of vertices. In light of this, the paper suggests approximating the weights in Delaunay Triangulation using a tight lower bound, which significantly accelerates the algorithm, making it practically linear. Additionally, the work introduces another per-vertex parameter that defines the probability of existence of each face that includes this vertex (given that it exists in WDT). Extensive validation shows that DMesh is capable of reconstructing complicated meshes, including open surfaces and non-orientable surfaces. Furthermore, DMesh outperforms the baselines on the tasks of surface reconstruction from a point cloud and multi-view rendering, while also being a more flexible and general representation.

**Strengths:**

The paper is easy to follow, and despite the complicated nature of the topic, the narration remains clear and formal; main concepts are well illustrated. The proposed modifications to Differentiable Surface Triangulation are significant and not straightforward. Extensive evaluation of the proposed method proves its effectiveness and significance for the field.

*Additional notes*:
+ Many regularization terms can be easily expressed natively for this representation, as demonstrated in the paper.
+ To the best of my knowledge, the paper compares against all the required baselines.
+ The supplementary material provides seemingly all the details needed to reproduce the algorithm.
+ The paper provides an efficient CUDA implementation of the algorithm (though I did not test it).
+ The demonstrated results are very impressive, and the resulting meshes look very clean.
+ The limitations of the paper are discussed in great detail.

Given all of the above, I believe this paper is well-positioned to significantly contribute to the community and potentially return attention to explicit representations in many fields, such as differentiable rendering, physics-based optimization, and so on.

**Weaknesses:**

I am a bit confused why the paper states that the running time of “Differentiable Surface Triangulation” growth exponentially with the number of vertices, while in fact it is quadratic? I would like to see either an explanation or a corrected version by the rebuttal.

While the method is very well evaluated and ablated, I think a few additional results will be interesting to see. Here they are in the order of decreasing significance:

1.	The biggest question to me so far is if the interpolations between two DMeshes are possible, especially if there is a topology/connectivity change. All the presented experiments showcase an optimization process that can start in a good initialization of the DMesh. My concern is that, after convergence the DMesh should get somehow more binary in $\Lambda_{\text{wdt}}$ and $\Lambda_{\text{real}}$, what will make it hard to take it out of this local minima in the further optimization process. By the rebuttal I would like to see a simple demonstration of a mesh-to-mesh optimization with a surface topology change, or a discussion of this in the limitation section if this is not trivial.

2.	In the limitations it is mentioned that sometimes the resulting mesh is non-manifold. It would be interesting to see some examples of failures and an estimation how often it happens and why? (at least I did not find this in the paper).

3.	(Minor) The point cloud reconstruction results are demonstrated for 100k sampled points, which is very dense. It would be interesting to see how the quality of mesh reconstruction depends on the number of sampled points.

4.	(Minor) Table 2 is evaluated only on 11 meshes, which is not much. Would be great to see a bigger test set by the camera-ready deadline.

*I would be ready to even further increase my rating if these questions are addressed.*

Minor Notes and Typos:

- Line 91: “it is not ..”  seems to miss something
- Would be great to increase the resolution of illustrations. In particular, the pixelation is very visible on Figure 6., at least in my PDF viewer
- Footnote 3 on page 5 – seems to be a typo around “and p_j and p_k”

**Questions:**

Here I have a few questions that don't affect the rating, but are rather out of curiosity. Including them in the final manuscript is up to the authors:
1. It looks like the representation allows the user to freeze the connectivity of the mesh if they want to. Is it also possible to "freeze" surface topology ?
2. Did you have to prevent the optimization process from converging to binary values of the probability ? (which I believe will make the mesh effectively non-differentiable ?)
3. is there anything we can say about the existence and uniqueness of the additional two parameters ($\Lambda_{\text{wdt}}$ and $\Lambda_{\text{real}}$) when reconstructing a mesh with the reconstruction loss from section 4.3.1. ?

After the authors addressed the questions in the rebuttal, increasing my rating to strong accept.

**Limitations:**

The limitations are discussed in great detail, apart from the lack of examples of non-manifold meshes as mentioned in Weaknesses. The broader impact discussion does not seem to be necessary here.

---

> ### Author Rebuttal · Authors · 2024-08-05
>
> We appreciate your detailed comments and positive evaluation about our work. Please let us know if we fully address your questions.
>
> ------------------------
> **Q1. Description about running time of “Differentiable Surface Triangulation”**
>
> A1. Thank you for pointing out the error. We acknowledge that our description was incorrect. We will change the description to "quadratic" instead of "exponential" in the revised version.
>
> -------------------------
> **Q2. Interpolations between two DMeshes, especially with topology/connectivity changes**
>
> A2. Please see Global Response A3. Additionally, please refer to Q8 below to understand how we prevent our formulation from becoming too binary or non-differentiable.
>
> -------------------------
> **Q3. Examples of non-manifold meshes, estimation of frequency, and reasons for occurrence**
>
> A3. Please see Global Response A2.
>
> -------------------------
> **Q4. Point cloud reconstruction results with varying densities**
>
> A4. Thank you for the suggestion. We will include experimental results in the final version of the paper to demonstrate how the quality of mesh reconstruction depends on the number of sampled points. We expect the results to degrade as the number of sampled points decreases since we are optimizing only Chamfer distance loss without any topological assumptions. Developing a more robust reconstruction algorithm based on our method would be an exciting future research direction.
>
> -------------------------
> **Q5. Evaluation on a larger test set**
>
> A5. Thank you for the suggestion. We will include more examples from the Objaverse dataset in the final version of our paper.
>
> -------------------------
> **Q6. Minor typos**
>
> A6. Thank you for identifying these. We will fix them in the revised version.
>
> -------------------------
> **Q7. Freezing surface topology**
>
> A7. As we understand, surface topology corresponds to geometric topology (e.g., genus of the shape) rather than mesh topology (individual edge connectivity) (Appendix A). If this is the case, we can freeze the surface topology by freezing all mesh connectivity, as you suggested. However, if you refer to *changing mesh connectivity while freezing surface topology, our method does not support this functionality yet*, because our method does not make any specific assumptions about surface topology during optimization. This would be an interesting research direction to explore. Please let us know if we understood this question correctly.
>
> -------------------------
> **Q8. Preventing optimization from converging to binary values of probability**
>
> A8. To answer this question, let us consider two cases where the existence probability of a face converges to either 1 or 0.
>
> First, if the probability has converged to 1, but we want to decrease it, we can simply adjust the real values stored in the 3 vertices of the face to reduce its probability. Since $\Lambda_{real}$ does not include any nonlinear function like sigmoid, we do not worry about falling into a local minimum due to the vanishing gradient problem.
>
> Next, let us assume the probability has converged to 0, but we want to increase it. There are two possibilities for the probability converging to 0: $\Lambda_{wdt}$ becoming 0 or $\Lambda_{real}$ becoming 0. If $\Lambda_{wdt}$ is almost 0, we did not use any measures because there are many possible combinations with very small $\Lambda_{wdt}$. Enumerating all possible face combinations (up to $n \choose 3$) and computing $\Lambda_{wdt}$ for them requires significant computational cost, making it impractical to amend this problem efficiently. However, using WDT is beneficial because the face existence probability usually increases as its vertices become closer. Thus, we observed that reconstruction loss moves points to minimize the loss, usually increasing (formerly very small) $\Lambda_{wdt}$ of the desirable face.
>
> If the probability of a face converges to 0 because $\Lambda_{real}$ is 0, we need a measure to prevent it because we ignore faces with very small probability to reduce the computational burden (Appendix C.3, C.5). These faces cannot get gradients to escape zero probability without any measures. Hence, we introduced the real regularizer in Appendix C.5. The regularizer increases the real value of points connected to points with high real values, helping escape local minima effectively.
>
> We will include this detailed discussion about local minima in the revised version of the paper.
>
> -------------------------
> **Q9. Existence and uniqueness of $\Lambda_{wdt}$ and $\Lambda_{real}$.**
>
> A9. Thank you for the question. We believe $\Lambda_{wdt}$ is related to the tessellation of the spatial domain, and $\Lambda_{real}$ is about selecting desired faces from the tessellation. Thus, $\Lambda_{real}$  depends on $\Lambda_{wdt}$: if tessellation changes, selected faces may change accordingly. When $\Lambda_{wdt}$ is fixed (and thus tessellation is fixed), we could argue that $\Lambda_{real}$ that globally minimizes the reconstruction loss would exist and be unique.
>
> Regarding $\Lambda_{wdt}$, we refer to literature on WDT [1]. Certain constraints, like those related to Local Feature Size (LFS), guarantee we can recover a topologically equivalent object to the ground truth mesh by selecting faces from (W)DT. We believe this applies to our case and guarantees the existence of a solution theoretically. However, for uniqueness, we do not think only one $\Lambda_{wdt}$ reconstructs the given geometry. For instance, we can subdivide faces on the current DMesh by inserting additional points or end up with different $\Lambda_{wdt}$ due to different initialization. Thus, we believe we can theoretically guarantee the existence of $\Lambda_{wdt}$ but not its uniqueness.
>
> [1] Cheng, Siu-Wing, et al. Delaunay mesh generation. Boca Raton: CRC Press, 2013.

---

> ### Comment · Reviewer_yemP · 2024-08-08
>
> Thank you for the detailed response.
>
> For Q8:
> 1. when you say "we can simply adjust the real values stored in the 3 vertices", do you mean that the optimization process naturally has gradients that minimize $\Lambda_\text{real}$ or that you need to embed this behavior separately?
> 2. In the case of the probability of face existing = 1, are there any cases when we would prefer the optimization process to decrease $\Lambda_\text{wdt}$ instead? I am curious to understand if the over-parametrization caused by having two parameters ($\Lambda_\text{real}$ and $\Lambda_\text{wdt}$) might cause any kind of dead-locks in the optimization process. As you mentioned in Q9, for a fixed tesselation there is a global minimizer in terms of $\Lambda_\text{real}$, but I believe that to efficiently optimize from one surface to another we will need to change tessellation first and only then optimize $\Lambda_\text{wdt}$.
> 3. Thank you for providing the mesh-to-mesh optimization experiment. This is very interesting. Do I understand correctly that the optimization loss was defined through multi-view inverse rendering? I noticed that the tessellation has changed and that in both cases the meshes seem to get bigger triangle size. Do you have any intuition why this happens?

---

> ### Author Response · Authors · 2024-08-09
>
> Thank you for the great questions, it inspires us a lot. To answer your additional questions,
>
> ---
>
> 1. We have not embedded any specific behavior to prefer $\Lambda_{real}$ over $\Lambda_{wdt}$, or manually optimize $\Lambda_{real}$ instead of $\Lambda_{wdt}$ in the optimization process. Technically, $\Lambda_{real} = 1$ and $\Lambda_{wdt} = 1$ would get the same amount of gradient, because they are just multiplied together to get the final face existence probability (Eq. 2). However, note that $\Lambda_{wdt}$ includes sigmoid function in its formulation (Eq. 4). Therefore, the parameters that are fed into the sigmoid function could get very small gradient when $\Lambda_{wdt}$ is almost binary. In contrast, $\Lambda_{real}$ does not have sigmoid function in its formulation, which means that we do not have to worry about vanishing gradient problem for $\Lambda_{real}$. In this case, the optimization process would "naturally" optimize $\Lambda_{real}$ instead of $\Lambda_{wdt}$, because $\Lambda_{wdt}$ would not be very optimizable. So we intended to say that even when $\Lambda_{wdt}$ is binary, optimizer would be able to adjust $\Lambda_{real}$ to escape the binary state.
>
> ---
>
> 2. As described above, if $\Lambda_{wdt}$ is almost binary, optimizer would automatically adjust $\Lambda_{real}$ instead of $\Lambda_{wdt}$. And as far as we understand, your concern is about only manipulating $\Lambda_{real}$ instead of $\Lambda_{wdt}$, even when we could get better optimization result when we change $\Lambda_{wdt}$ instead to get better tessellation. Correct us if we are wrong, but if our understanding is correct, our current approach does not care about this possibility yet. Our current algorithm minimizes the loss function in blind manner right now, regardless of specific $\Lambda_{real}$ and $\Lambda_{wdt}$. Therefore, we would have to say it finds the local minimum near current tessellation, instead of finding the global minimum near the optimal tessellation. However, we believe that the coarse-to-fine strategy in Appendix D.2 is a very good method to re-initialize $\Lambda_{wdt}$, thus tessellation, near the global optimal tessellation. Therefore, we'd like to suggest to use that method to find the global minimum.
>
> ---
>
> 3. Yes, as you mentioned, the optimization loss was defined with multi-view inverse rendering. The reason about the triangle size is related to the answers above. Because $\Lambda_{real}$ is usually easier to optimize than $\Lambda_{wdt}$, it is optimized faster. That is, while the entire tessellation does not change a lot, only the faces that we select change fast.
>
> To elaborate, at the end of the first optimization to fit DMesh to a single torus, most of the points are gathered around the torus surface. Since there are not so many points located at the places far away from the torus surface (as we remove unnecessary points using regularization), the faces in that region become larger than those located near the torus surface. Then, we start from that DMesh to fit different shapes like double torus, and in that process, rather than moving (single) torus surface toward the target shape, the optimizer just removes the undesirable parts on the torus surface by adjusting $\Lambda_{real}$. And then, it makes the probability of desirable faces larger, and they usually have larger faces, because they were located far away from the original single torus. After that, $\Lambda_{wdt}$ is optimized slowly to fine-tune the shape.
>
> After this process, if we re-initialize DMesh using the coarse-to-fine strategy, then it would get much better initial tessellation to start from. Then, it would end up in much smaller, fine-grained faces in the end.
>
> We believe it would be very interesting to explore if we can impose any kind of guidance to select (or prefer) between $\Lambda_{wdt}$ and $\Lambda_{real}$ during the optimization. Thank you again for the inspiration.

---

> ### Comment · Reviewer_yemP · 2024-08-09
>
> Thank you for the fast reply. This is very interesting. It will be interesting to look if stopgradient or just rescaling the gradients will help to define the balance between $\Lambda_\text{wdt}$ and $\Lambda_\text{real}$.
>
> I think the mesh-to-mesh example will be a good addition to support the claims of the paper, so please consider including this in the final version.
>
> At this point, I don't have any more questions and will gladly adjust my recommendation.

---

> > ### Author Response · Authors · 2024-08-09
> >
> > Thank you very much, we agree that we can consider such balancing techniques!
> >
> > Also, we will definitely add the additional experimental results in the paper, thank you for the suggestion.
> >
> > It would be really big help for us if you could give a raise to our rating, we will really appreciate it :)

---

### Official Review · Reviewer_cnRb · 2024-07-14

**Soundness:** 2
**Presentation:** 3
**Contribution:** 2
**Rating:** 6
**Confidence:** 4

**Summary:**

The authors first propose a differentiable mesh representation, DMesh, which can represent a wider variety of mesh types. They then introduce a computationally efficient approach to differentiable weighted Delaunay triangulation that can run in approximately linear time. Following this, an efficient algorithm is provided for reconstructing surfaces from both point clouds and multi-view images, with DMesh used as an intermediate representation. Finally, an effective regularization term is proposed for mesh simplification and enhancing triangle quality.

**Strengths:**

The method proposed by this paper aims to optimize the geometry and topology of the reconstructed mesh. It presents a computationally efficient approach to differentiable weighted Delaunay triangulation, which can achieve this goal. The methods in this work are partly valuable to the NeurIPS community.

**Weaknesses:**

In this work, the inner structure is removed via a depth test, meaning that any reconstruction method producing an inner structure can be improved by using the depth test to remove the unseen parts of the mesh. While this post-processing step does help to improve the quality of the result, it is not a feature of the proposed method itself.

**Questions:**

1. During the post-processing step, the inner structures are removed via depth testing. How does the method handle the boundaries caused by the previous operation?

2. In line 91, "... and it is not." should be completed for clarity. Maybe the authors missed some words at the end.

3. The DMesh is constructed from the normal mesh. However, in real applications we often do not have access to the mesh. How does the DMesh help in reconstructing the result?

**Limitations:**

Yes.

---

> ### Author Rebuttal · Authors · 2024-08-05
>
> We appreciate your comments, but we believe there may have been some major misunderstandings about our work. We have tried to address your comments as thoroughly as possible. If we have not correctly addressed your concerns, please let us know.
>
> ---------------------------
>
> **Q1. Weakness about “depth testing”**
>
> A1. We believe there was a misunderstanding regarding the term "depth test" in our manuscript. We used this term to *describe the differences between the two renderers we employed in the multi-view reconstruction task* (Appendix C.3.1, C.3.2).
>
> In estimating the relative ordering between faces, the first renderer $F_A$ (Appendix C.3.1) can produce incorrect orderings because we compute the depth value of each face globally, which might not align with local orderings. This could result in inner faces being perceived as “seen” from the outside, leading to false inner structures (Figure 11(a)). This renderer is used in the first phase of optimization.
>
> In contrast, the second renderer $F_B$ (Appendix C.3.2) produces precise orderings between faces, thereby removing these false inner structures (Figure 11(b)). This renderer is used in the second phase of optimization, not during post-processing.
>
> In the post-processing step, we remove unseen inner faces based on the observation inherent to our approach (Appendix D.2.7), which tessellates the entire domain. Since other methods do not use this kind of tessellation, this approach is not applicable to them.
>
> We acknowledge that our use of the term could have been misleading. However, as described above, *it differs from conventional "depth test" or "visibility test" techniques that can be applied to any reconstruction method*. We used the term to *describe a situation inherent to our approach, not a general case*. *We did not perform any conventional "depth test" or "visibility test" to remove inner structures*. We will clarify this point in the revised version.
>
> -------------------------
> **Q2. During the post-processing step, the inner structures are removed via depth testing. How does the method handle the boundaries caused by the previous operation?**
>
> A2. As described above, the post-processing step in our method is based on principles inherent to our approach, not a general depth testing. Therefore, it does not produce additional boundaries on its own or get affected by boundaries produced in previous optimization phases.
>
> -------------------------
> **Q3. In line 91, "... and it is not." should be completed for clarity. Maybe the authors missed some words at the end.**
>
> A3. Thank you for pointing this out. We intended to write: "... and is only confined to handle point clouds.” We will correct this in the revised version of the paper.
>
>
> -------------------------
> **Q4. The DMesh is constructed from the normal mesh. However, in real applications, we often do not have access to the mesh. How does the DMesh help in reconstructing the result?**
>
> A4. When we reconstruct a mesh from point clouds or multi-view images, **we do not use the normal (ground truth) mesh as input**. In our experiments, we sample point clouds or capture multi-view images of the ground truth mesh, which are then fed into the reconstruction algorithm. *The ground truth mesh is used only to evaluate the reconstruction quality*. Thus, *our reconstruction algorithm can be used even when there is no normal mesh available*. Please also refer to the videos about point cloud and multi-view reconstructions in the index.html file in our supplementary material. These videos demonstrate that our method does not require the normal mesh as input.

---

> > ### Comment · Reviewer_cnRb · 2024-08-13
> >
> > Thanks for the authors' response with details, it solved my questions and I'd like to change my rating to weak accept.

---

> > > ### Author Response · Authors · 2024-08-13
> > >
> > > We're happy to hear that our response solved the questions, thank you for the raise!

---

### Author Rebuttal · Authors · 2024-08-05

Thank you for all of the reviews. We can further improve this paper based on your feedback.

Before addressing the specific points, *we encourage the reviewers to refer to the index.html file in the supplementary material, which contains videos visualizing the meshes during optimization in our experiments*.

---

**Q1. Details about computational cost (Reviewer 8dk4, jRqE, 7cDg)**

A1. We conducted additional experiments to analyze computational costs, particularly time. **Figure 1** in the attached PDF shows how computational time changes with the number of points.

We classified computational time into WDT (computing WDT and PD of given points) and Prob (computing face probabilities based on WDT and PD). Refer to Algorithm 2 in Appendix D for details. For each point count, we generated 3D point clouds in a unit cube with uniform weight 0 and ran each setting 5 times to get average running times.

As shown in the graph, computational cost increases with the number of points due to an increase in possible faces, thus increasing Prob's cost. However, *WDT takes most of the time, validating our claim that WDT is the main computational bottleneck in the current implementation*. Although the cost rises rapidly beyond 20K points, the algorithm is affordable below 20k and efficient below 10K points. In fact, most single objects can be represented with ~4K points, as shown in Table 2 in paper. *Therefore, our approach is efficient for most single objects*. However, larger scenes require more points, potentially up to millions. We are exploring an accelerated implementation of DMesh that efficiently handle millions of points.

---

**Q2. Details about non-manifoldness (Reviewer yemP, 7cDg)**

A2. To explain cases where the resulting mesh is non-manifold, let's revisit the combinatorial definition of manifoldness. First, an edge should be incident to at most 2 faces. An edge incident to more than 2 faces is a non-manifold edge. Second, faces incident to a vertex should form a closed or open fan. A vertex not meeting this condition is a non-manifold vertex. Finally, there should be no self-intersections between faces. Our method guarantees no self-intersections, so our measurement focuses on the first two conditions. Testing on 11 manifold models, we measure the average ratio of non-manifold edges and vertices.

For point cloud reconstruction results, 5.50% of edges were non-manifold, and 0.38% of vertices were non-manifold. For multi-view reconstruction results, 6.62% of edges were non-manifold, and 0.25% of vertices were non-manifold. Therefore, *non-manifold edges are more significant than non-manifold vertices in our approach*. The main reason for non-manifold edges is a corner case in our formulation. We select faces using point-wise real values from each tetrahedron in the tessellation, but *if we need to select 2 or 3 faces, all points should have a real value of 1, resulting in selecting all faces that could produce non-manifold edges*. Please refer to **Figure 2** in the attached PDF for illustrations. Because of this ambiguity, *redundant faces are created and non-manifold edges arise from them*. We believe *optimizing face-wise real values in post-processing can considerably reduce non-manifold edges*, thereby avoiding this corner case.

Despite not preventing non-manifold cases, we see this as a *double-edged sword regarding representation capacity*.  Enforcing manifold constraints is beneficial for representing real-world objects, BUT less so for abstract geometries, where non-manifold cases often arise, e.g., the plant and great stellated dodecahedron in the teaser and Bigvegas in Appendix E. In fact, many popular 3D assets, such as Mixamo, contain non-manifold meshes for cloth and surfaces. *Therefore, we purposely design our method to be a versatile geometric representation that encompasses all these cases*. Exploring constraints to guarantee *user-desired level of manifoldness* using our proposed representation is a promising future research direction for application-specific use and we would be happy to highlight it in the revision.

---

**Q3. Interpolations between DMesh exhibiting topology / connectivity changes during deformation (Reviewer yemP, 7cDg)**

A3. Firstly, we would like to emphasize that the videos embedded in the index.html file already demonstrate how topology / connectivity changes during optimization. We would appreciate it if reviewers could please review these supplementary videos.

We agree that it would be beneficial to see interpolations between DMesh of different (surface) topologies, especially starting from a converged DMesh due to local minima issues. To illustrate this, we first fitted our DMesh to a **single torus** (torus with 1 hole), and then optimized it to 2 other shapes, which include a **similar shape with the same genus**, and **double torus** (torus with 2 holes), assuming the multi-view images are provided as input. Please refer to **Figure 3** in the attached PDF, which shows the optimization process.

As demonstrated in Figure 3, *our method successfully recovers the target shape, even when starting from an already converged single torus shape*. Therefore, we argue that *concerns about local minima may not be as significant*. This also highlights *our approach's capability in optimizing topology explicitly*. However, we acknowledge that more complex cases might exist. For such cases, we propose using our *coarse-to-fine strategy* (Appendix D.2), to *effectively escape (possible) local minima*.  Therefore, we claim that *our method can robustly address local minima issues*.

However, in Figure 3, we observe that the intermediate states between the two shapes are not semantically meaningful. Even when we optimize for a shape of same surface topology (blue), it does not go through a smooth interpolation. This is because *we do not have any topological assumption*, which is double-edged sword of our formulation, as discussed in above A2.

---

### Author Response · Authors · 2024-08-07
**About global response**

Sorry for your inconvenience, but we just found out that the global response is not visible to the reviewers.

We will make it visible to reviewers as soon as possible.

--------- update

It is visible now, thank you.

---

> ### Comment · Area_Chair_XUj8 · 2024-08-07
>
> It seems to me that your global answer is already visible to reviewers.
>
> AC

---

> > ### Author Response · Authors · 2024-08-07
> >
> > Oh, it is visible now. Thank you very much!

---

### Decision · Program_Chairs · 2024-09-25

**Decision:**

Accept (poster)

**Comment:**

This paper introduces a differentiable representation for meshes and efficient algorithms supporting it, allowing surface reconstruction. The paper has received favorable reviews from five expert reviewers, who generally appreciated the idea, efficiency (including a CUDA implementation), good presentation and writing, extensive evaluations (including appendix the paper is 42 pages long ...), extensive comparisons, discussed limitations, etc.

Some weaknesses were raised (manifoldness, remaining computational cost), which did not impact the suitability for publication.
The AC concurs and judges the paper to be strong and ready for publication.